# Pandemic preparedness in shaping psychosocial working conditions – insights for occupational safety and health from a longitudinal mixed-methods study during the COVID-19 pandemic at six company sites of one organization in Germany

Jana Soeder[ID][1]*, Christine Preiser[1], Anke Wagner[1], Anna T. Neunhöffer[1], Falko Papenfuss[2], Juliane Schwille-Kiuntke[1], Andrea Wittich[3], Esther Rind[1☙], Monika A. Rieger[ID][1☙]

1 Institute of Occupational and Social Medicine and Health Services Research, University Hospital Tübingen, Tübingen, Germany, 2 Medical Services, Robert Bosch GmbH, Stuttgart, Germany, 3 Occupational Psychologist and Psychotherapist, Tübingen, Germany

☙ These authors share last authorship.
* jana.soeder@med.uni-tuebingen.de

## Abstract

### Objectives

This longitudinal mixed-methods study explores how one company group dealt with COVID-19 pandemic-related challenges and how employees and managers perceived work-related psychosocial demands in Germany.

### Materials and methods

We analyzed panel survey data from 322 employees and managers across diverse working fields, e.g., assembly line/production, office, company medical service, and factory security service. Employees and managers were recruited from six German company sites in the federal states of Bavaria, Baden-Wurttemberg, and Lower Saxony of one large company group. The survey was conducted in August-October 2020, January 2021, and October-November 2021. Participants self-reported their perceived psychosocial demands from aspects of work organization, work environment, work content, and social relations at work during and retrospectively before the COVID-19 pandemic on a 5-point Likert scale. Additionally, nine managers were interviewed in September-October 2020 and April-May 2021 about pandemic-related changes in working conditions, organizational processes to adapt working conditions, and a culture of trust. For contextualization, we performed a comprehensive document analysis of prevailing national and federal laws and OSH-regulations for infection control in Germany.

**Data availability statement:** We understand the importance of data sharing in ensuring transparency and replicability of research. However, due to our ethical commitments to the participating company and their employees, we stored the data collected in a protected electronic database within the network of the University Hospital, Tübingen. This database can only be accessed by authorized research employees of the Institute of Occupational Medicine and Social Medicine and Health Services Research, University Hospital Tübingen. Ethical approval was obtained by the Medical Faculty, the University of Tübingen, and the University Hospital of Tübingen in June 2020 (No.: 423/2020BO). The address of the committee that validated the ethical and methodological protocol for this research is as follows: Ethics Committee at the Medical Faculty of the Eberhard Karls University and at the University Hospital of Tübingen, Gartenstraße 47, 72074 Tübingen, Geschäftsstelle, Tel. +49 (0)7071 29-77661, E-Mail: ethik.kommission@med.uni-tuebingen.de. On reasonable request, access to the dataset should be directed to the first author Jana Soeder (jana.soeder@med.uni-tuebingen.de), or the project team leaders Esther Rind (esther.rind@med.uni-tuebingen.de) and Monika A. Rieger (monika.rieger@med.uni-tuebingen. dearbeitsmedizin@med.uni-tuebingen.de).

**Funding:** The overall research project was funded by the Ministry of Science, Research and Art, Baden-Württemberg. The work of the Institute of Occupational and Social Medicine and Health Services Research Tübingen is supported by an unrestricted grant of the employers´ association of the metal and electric industry Baden-Württemberg (Südwestmetall). The funding bodies had no role in the design of this study nor during its execution, analyses, interpretation of the data, or in the decision to publish the results. We gratefully acknowledge support from the Open Access Publishing Fund of the University of Tübingen.

**Competing interests:** I have read the journal's policy and the authors of this manuscript

## Results

Quantitative results revealed that participants perceived stable psychosocial demands during the pandemic. Psychosocial demands relating to work organization were perceived as stressful, especially for managers and company's medical service personnel. No statistically significant changes were identified with respect to psychosocial demand ratings during compared to before the pandemic. Qualitative results highlighted that a crisis management team, a culture of trust, extensive and transparent communication, and participatory approaches in change processes were key to managing pandemic-related challenges.

## Conclusion

Our findings contribute to a contextualized understanding of how the design of good and sustainable working conditions ensuredwork ability during the COVID-19 pandemic in Germany. They highlight the importance of pandemic preparedness in an organization's psychosocial work design to successfully manage challenging times. The derived learnings for organizational pandemic preparedness on a meso-, macro-, and micro-level are of lasting significance with a view to future global challenges in terms of the design of good psychosocial working conditions.

## Introduction

The International Labor Organization (ILO) and Organization for Economic Co-operation and Development (OECD) highlight that digitalization, ecological transformation, and advancing demographic change are reshaping the world of work [1]. With regard to opportunities for decent work within the G7 nations, they jointly state that these changes in working conditions, e.g., extended periods of remote work, impact psychosocial demands [1]. Psychosocial risk factors, including long working hours and employment uncertainty, are strongly associated with health issues such as burnout, coronary heart disease, and anxiety [2,3]. In 1989, the European Union adopted the European Framework Directive 89/391/EEC-OSH on Safety and Health at Work [4]. The directive was meant to emphasize an employer's responsibility of ensuring employee occupational safety and health (OSH). Member states of the European Union transposed this directive into their country-specific structures and national legislation [5]. In Germany, OSH is regulated by the Safety and Health at Work Act (ArbSchG) [6]. Here, a risk assessment is the first step in deriving and implementing OSH-measures aimed at preventing work-related accidents and health hazards. Possible work-related risks can be elicited by both, physical and psychosocial demands. Psychosocial demands can – depending on the individual's preconditions such as socio-demographic characteristics, experiences, and current mental state – lead to either favorable (e.g., activating state) or unfavorable strain (e.g., fatigue or problems concentrating) and thus impact the individual's health [7]. Psychosocial demands in the workplace refer to key aspects of work organization,

have the following competing interests: the authors JS, CP, AW, AWi, ATN, ER, MAR declare no conflicts of interest. JS-K's sole permanent employment relationship is with the Regierungspräsidium Tübingen/public health department Hechingen, Zollernalbkreis. She declares no conflict of interest. FP has been involved as consultant, expert, and co-author, and is employed at the Robert Bosch GmbH. FP has been primarily involved in developing the study idea and the design and content of the online employee survey. The participating company had no role in the analysis of data, the interpretation of results, or in the decision to publish the results.

work environment, work content, social relations, and new forms of work. An aspect of work organization, for example, is the constant interruption during work, while an aspect of work content is the degree of influence a worker has over assigned tasks. These aspects are to be considered when conducting a psychosocial risk assessment in the workplace, according to the Joint German OSH-Strategy (GDA) [7].

Systematic reviews show that psychological distress increased and the health of individuals was negatively affected during previous crises, e.g., Ebola epidemic or the economic recession of 2008 [8,9]. Exacerbating factors included being unemployed, job loss, income loss, precarious work, and pre-existing mental disorders. The COVID-19 pandemic's unpredictability made it challenging for companies to maintain economic productivity. Comprehensive measures were introduced to prevent SARS-CoV-2 infections at work, outbreaks at work spreading to families, a high rate of sick leave, and overwhelmed healthcare systems. As a consequence, workplaces were adapted: for example, re-arrangement to maintain a distance of 1.5m between workstations (technical), remote work where possible (organizational), and the wearing of face masks if distance could not be maintained (personal measures). Thus, daily working conditions changed rapidly as a result of the implemented SARS-CoV-2-infection control measures. Physical distancing on-site, increased remote work, and new cleaning and hygiene protocols led to difficulties in communication flows, blurred boundaries between work and private life, and unclear responsibilities. The dynamic pandemic situation required ongoing re-adaptations in OSH-measures. Those continuous changes in how and under which conditions people work, impacted the work-related psychosocial demands employees experienced during the pandemic. A longitudinal employee survey among employees in Germany revealed that organizations can play an important role in supporting employees to accept these new working conditions [10]. Providing good information about possible work-related risks and designing the work environment appropriately, for example, seemed to support employees in maintaining long-term favorable attitudes toward health-related preventive behaviors during the COVID-19 pandemic [10]. When it came to re-designing working conditions during the pandemic, telephone interviews revealed that managers played a key role in German companies [11]. Ideally, they had the flexibility to adapt tasks or working times to the specific needs of employees. Findings from the European Working Conditions Survey revealed differences between working fields with respect to new working conditions and the challenges associated with them [12]. Those who were able to work from home usually faced long working hours while on-site production workers often experienced job insecurity and income loss. Employees were found to trust managerial decisions more when they were involved in organizational decision-making [12].

Despite this solid body of evidence, a knowledge gap exists on how organizational responses to the challenges resulting from the COVID-19 pandemic relate to the employees' perceived psychosocial demands at work. Organizational responses include, for example, communication strategies, approaches to re-designing on-site work environments, and the adaptation of infection control measures tailored to specific working fields. Therefore, the question is, how organizations addressed

pandemic-related challenges, focusing on working conditions. It is of further interest, how employees and managers perceived psychosocial demands during and retrospectively before the COVID-19 pandemic in Germany.

## Materials and methods

### Study design

This study is part of a larger, exploratory, modular study project that examines the acceptance of SARS-CoV-2-infection control measures in occupational settings [13,14]. Ethical approval was obtained by the Medical Faculty, the University of Tübingen, and the University Hospital of Tübingen in June 2020 (No.: 423/2020BO). The study project was conducted in accordance with the Declaration of Helsinki and relevant institutional guidelines.

For the current study, we applied a mixed-methods approach, commonly used in organizational health services research to consider the complexity of organizations [15,16]. Using the combination of qualitative interviews and a quantitative survey, we aimed to gain insights into organizational pandemic preparedness in work design. Quantitative survey data were collected at multiple timepoints to monitor changes in how the employees' and managers' perceived psychosocial demands at work. Findings from the interviews further helped to address how workers of different working fields experienced support from both the German state (e.g., financial aid in the form of temporary short-time work compensation) and their organization (e.g., adjusted work arrangements) during the pandemic. These pluralistic perspectives enable an evaluation of how the organization's responses to the challenges of the pandemic affected the employees' daily work.

The findings from this study could help design more favorable and sustainable psychosocial working conditions in organizations that are similar to the one investigated. Designing working conditions based on the learnings from the COVID-19 pandemic can help ensure work ability during future challenging times.

### Mixed-methods approach

For contextualization, we performed a comprehensive document analysis of prevailing national and federal laws and OSH-regulations for infection control in Germany. In a parallel approach, quantitative (employee survey) and qualitative data (interviews with managers) were gathered and analyzed independently. This allowed us to gain detailed insights into the company-level context in which employees and managers work. Data integration was realized during sampling, as both, quantitative and qualitative data, were collected via convenience sampling from the same working fields to ensure comparability. Further, integration was realized during the exploratory analysis and interpretation phase. The triangulation of methods allowed us to investigate consistent patterns in perceived psychosocial demands within working fields as well as between employees and managers. The contextual insights, mainly provided by the qualitative data, allowed a more nuanced understanding of these patterns. We report following the GRAMMS-guidelines [17].

### Study participants

The longitudinal data comes from employees and managers from various working fields from a worldwide leading global supplier of technology and services. The company group can be characterized as financially well-established and having a lot of human resources [10]. For example, regarding the work environment, digitalized solutions and proven communication channels were already used before the COVID-19 pandemic. All employees and managers included were employed at six German company sites in the federal states of Bavaria, Baden-Wurttemberg, and Lower Saxony. Employees for example worked at the assembly line/production, in the office, company medical service, or factory security service of the same company group. We aim to differentiate between those working fields as they show large differences in how working conditions were adapted to ensure as-safe-as-possible workplaces. While office workers were mainly affected by remote work and online meetings where possible, assembly line/manufacturing workers experienced redesigned working conditions to allow for keeping a minimum of 1.5m distance between work stations, frequent cleaning of work equipment, and

decoupled break times. Company medical service personnel in comparison were responsible for implementing SARS-CoV-2-infection control measures as well as ensuring testing and vaccination possibilities. All participants were at least 18 years old, had a German language proficiency of at least B1, and provided written informed consent.

Employees were eligible as interview partners if they had been employed for at least six months and were either member of the crisis management team or held an operational management position. No explicit exclusion criteria were defined beyond not meeting the pre-defined inclusion criteria. Due to the limited time frame between the onset of the pandemic and the first wave of data collection, the employee survey was administered solely online [18]. As a consequence, there may have been systematic exclusion of individuals without sufficient time or resources to participate such as internet or email access at work. This was previously discussed in Soeder et al. [18]. Qualitative data were pseudonymized and quantitative data were anonymized prior to analysis.

## Quantitative data collection and analysis

Quantitative self-reported employee survey data were generated by distributing an online survey three times after the onset of the COVID-19 pandemic in Germany: August 10 to October 25, 2020 (T0), January 21 to 31, 2021 (T1), and October 15 to November 21, 2021 (T2). The questionnaire was designed using the online survey tool Unipark provided by Questback AS [19]. Survey invitations including an access link and QR code were sent to all employees by the company's corporate communications department via email, newsletter, intranet, postcards, and posters, see Soeder et al. [18] for more details. Participation was voluntary and we used convenience sampling. Multiple observations were linked to the same individual using self-generated anonymized codes. Details on recruiting strategies, surveyed items, and response rate at T0 (22%) are described elsewhere [18].

Primary outcome variables are the perceived psychosocial demands in the workplace, see the S1 Table. For each item, we asked for evaluation from two perspectives: retrospectively before and currently during the COVID-19 pandemic. The survey items were self-developed following the GDA recommendations [7] and Copenhagen Psychosocial Questionnaire (COPSOQ) [20]. Each item was to be rated on a 5-point Likert scale, symmetrically anchored with '3', which indicates a neutral perception. Some items were reversed for score computation, see S1 Table for more details. Items were grouped into four scores (three items each) by four experts in occupational medicine and health services research: perceived psychosocial demands from aspects of the work organization, work environment, work content, and social relations at work. Scores were computed following the mean-across-available-item approach (S2 Table) [21]. High values reflect favorable, low values reflect unfavorable perceptions of work-related psychosocial demands. Additional variables included in this analysis were: working field, remote work at the timepoint of the survey (yes/no), occupying a manager position (yes/no), and short-time work (yes/no).

We present the results with number to treat, mean±standard deviation (SD), and median and interquartile range (IQR) for continuous measures including Likert scales; absolute and relative frequencies for categorical variables. Outcome mean scores (work organization, environment, content, and social relations) were non-normally distributed and left-skewed. Friedman-Test was used to assess differences between timepoints. Wilcoxon's signed-rank test with Bonferroni correction was used for pairwise comparison. We report effect sizes with $r < 0.3$ (small), $r \geq 0.3$ and $r \leq 0.5$ (moderate), and $r > 0.5$ (strong effect) [22]. Wilcoxon rank-sum test was used to compare two groups, e.g., managers vs. non-managers, at the same timepoint. To compare the 'retrospective' and 'current during the pandemic' ratings at each timepoint, we first computed the within-person differences and second categorized the differences into 'equal =0', 'now better than before >0', and 'now worse than before <0'. Due to the exploratory study design, claims of causality are not possible. Statistical significance was set at two-sided $p < 0.05$. Statistical analyses were performed using R version 4.2.1 using the packages psych, rstatix, and ggplot2 [23]. All packages used for statistical analyses are available on CRAN (Comprehensive R Archive Network, https://cran.r-project.org/web/packages/).

## Qualitative data collection and analysis

Semi-structured interviews were conducted from September 28 to October 19, 2020 (t0) and from April 19 to May 6, 2021 (t1) via video call or telephone (24−60 minutes) by author 2 (CP). The interview guides (S3 Table) were drafted by an experienced sociologist (CP) who received feedback from experts in occupational medicine, health services research, and general medicine [24]. We were interested in how work-related challenges elicited by the COVID-19 pandemic were handled in the collaborative company group and how managers experienced these responses [25]. Potential interview partners were suggested by the company's research partners based on a minimum employment time with the company of six months. Interview partners were eligible if they were either members of the crisis management team or holding an operational management position. Author CP arranged interview appointments. All interviews began with a brief intro-duction, information about data protection, and the interview structure. Audio files were transcribed verbatim by a profes-sional transcription agency (amanu GmbH, Germany). The qualitative data were organized using MAXQDA 2020 (VERBI Software, Germany) and analyzed via qualitative content analysis [26] to identify and structure the main topics and sub-topics of the data (S4 Table). Regarding data saturation, the interviews showed a strong information power [24]. For the first interviews (t0), nine main categories were derived deductively from the interview guide. A definition and exempli-fying quote were added to each category for clarification. Then, several interviews were coded. All sections belonging to each main category were paraphrased to build data-driven subcategories. The coding frame was additionally piloted for the respective interviews, finalized after minor adjustments, and applied to the remaining interviews. All steps were carried out by two researchers working independently and discussing their analyses continuously. For the second data collection period (t1), the coding frame was updated following the same procedure. New interview questions, e.g., concerning the topic of vaccinations, resulted in additional categories. Both coding frames were discussed with the overall research team.

To contrast the qualitative with quantitative findings during the analysis phase, we here focus on the following three categories: pandemic-related changes in working conditions, organizational processes for adapting working conditions, and culture of trust.

## Results

### Sample characteristics

The here presented survey results are based on a sub-sample of participants (n = 322 out of N = 5,554) for whom all three survey observations were available. A total of 5,554 employees participated in the online survey at one, two, or all three timepoints. For 322 out of them, longitudinal panel data could be ensured. This analysis focuses on panel data, as we were primarily interested in within-person changes in psychosocial demands. The outcome variables and socio-demographic characteristics did not differ substantially between participants who took part three times compared to the overall sample, see S5 Table for an additional sensitivity analysis. See [10], for a pooled analysis of all available repeated cross-sectional and longitudinal panel data investigating the employees' attitudes toward organizational infection control measures in the workplace. Individual and work-related sample characteristics are shown in Table 1.

Eight male and one female managers participated in the interviews. Of these, n = 7 interview partners participated in the initial and follow-up interviews. The following perspectives were presented: factory security service, company medical service, works council, technical operation manager, corporate financial management, human resources, and assembly line/ manufacturing. Employees worked at six company sites in three German federal states of one large company group. This allows differentiation between working fields even with small sample sizes and thereby covering possible differences in safety cultures or local pandemic situations.

### COVID-19 pandemic in Germany

Fig 1 provides an overview of the main findings of the document analysis together with the data collection phases [27–29]. For example, the German social security concept included statutory employment-protection systems such as continued

**Table 1. Sample characteristics.** Characteristics of the study participants (n = 322) in the repeated employee survey and their perceived psychosocial demands during and retrospectively before the pandemic.

| Timepoint | T0 | | | T1 | | | T2 | | |
|---|---|---|---|---|---|---|---|---|---|
| Characteristics | Median (IQR) | Mean (SD) | n (%) | Median (IQR) | Mean (SD) | n (%) | Median (IQR) | Mean (SD) | n (%) |
| **Age** | 47.0 (17.0) | 45.0 (10.8) | 321 (99.3) | 48.0 (17.0) | 46.0 (10.8) | 321 (99.3) | 48.0 (17.0) | 46.0 (10. | 321 (99.3) |
| 18-29 | | | 37 (11.5) | | | 30 (9.3) | | | 30 (9.3) |
| 30-39 | | | 62 (19.3) | | | 61 (18.9) | | | 61 (18.9) |
| 40-49 | | | 96 (29.8) | | | 92 (28.6) | | | 92 (28.6) |
| 50-59 | | | 105 (32.6) | | | 108 (33.5) | | | 108 (33.5) |
| 60-71 | | | 21 (6.5) | | | 30 (9.3) | | | 30 (9.3) |
| missing | | | 1 (0.3) | | | 1 (0.3) | | | 1 (0.3) |
| **Gender** [a] | | | | | | | | | |
| female | | | 119 (37.0) | | | 119 (37.0) | | | 119 (37.0) |
| male | | | 203 (63.0) | | | 203 (63.0) | | | 203 (63.0) |
| **Working fields** | | | | | | | | | |
| assembly line | | | 31 (9.6) | | | 28 (8.7) | | | 33 (10.2) |
| company medical service | | | 6 (1.9) | | | 6 (1.9) | | | 6 (1.9) |
| factory security service | | | 9 (2.8) | | | 7 (2.2) | | | 6 (1.9) |
| office | | | 151 (46.9) | | | 106 (32.9) | | | 147 (45.7) |
| office remote >=50% | | | 109 (33.9) | | | 161 (50.0) | | | 119 (37.0) |
| other, e.g., trainee | | | 16 (5.0) | | | 13 (4.0) | | | 11 (3.4) |
| missing | | | 0 (0) | | | 1 (0.3) | | | 0 (0) |
| **Manager position** | | | | | | | | | |
| no | | | 263 (81.7) | | | 262 (81.4) | | | 259 (80.4) |
| yes | | | 59 (18.3) | | | 58 (18.0) | | | 63 (19.6) |
| missing | | | 0 (0) | | | 2 (0.6) | | | 0 (0) |
| **Fixed term contract** | | | | | | | | | |
| no | | | 321 (99.7) | | | 317 (98.4) | | | 318 (98.8) |
| yes | | | 1 (0.3) | | | 1 (0.3) | | | 1 (0.9) |
| missing | | | 0 (0.0) | | | 4 (1.2) | | | 1 (0.3) |
| **Employment at company** | 20.0 (16.0) | 19.4 (10.2) | 321 (99.7) | 20.0 (17.5) | 19.7 (10.2) | 318 (98.8) | 21.0 (18.0) | 20.3 (10.2) | 320 (99.4) |
| less than two years | | | 11 (3.4) | | | 7 (2.2) | | | 4 (1.2) |
| at least two years | | | 310 (96.3) | | | 311 (96.6) | | | 316 (98.1) |
| missing | | | 1 (0.3) | | | 4 (1.2) | | | 2 (0.6) |
| **Short-time work** [b] | | | | | | | | | |
| no | | | 299 (92.9) | | | 309 (96.0) | | | 299 (92.9) |
| yes | | | 21 (6.5) | | | 13 (4.0) | | | 22 (6.8) |
| missing | | | 2 (0.6) | | | 0 (0) | | | 1 (0.3) |
| **Perceived psychosocial demands during the COVID-19 pandemic rated on a 5-point Likert scale** | | | | | | | | | |
| work organization | 2.7 (1.0) | 2.8 (0.9) | 320 (99.4) | 2.7 (1.3) | 2.8 (0.9) | 321 (99.7) | 2.7 (1.0) | 2.7 (0.9) | 321 (99.7) |
| work environment | 4.0 (1.0) | 4.0 (0.9) | 320 (99.4) | 4.3 (1.3) | 4.1 (0.9) | 321 (99.7) | 4.3 (1.3) | 4.1 (0.9) | 321 (99.7) |
| work content | 3.7 (1.0) | 3.6 (0.8) | 320 (99.4) | 3.7 (1.0) | 3.6 (0.8) | 321 (99.7) | 3.7 (1.0) | 3.6 (0.8) | 321 (99.7) |
| social relation | 4.0 (1.0) | 4.0 (0.8) | 321 (99.7) | 4.0 (1.0) | 4.0 (0.8) | 321 (99.7) | 4.0 (1.0) | 3.9 (0.8) | 322 (100) |
| **Perceived psychosocial demands before the COVID-19 pandemic rated on a 5-point Likert scale** | | | | | | | | | |
| work organization | 2.7 (1.0) | 2.7 (0.8) | 320 (99.4) | 2.7 (1.0) | 2.7 (0.8) | 321 (99.7) | 2.7 (1.0) | 2.6 (0.8) | 320 (99.4) |
| work environment | 4.0 (1.1) | 4.1 (0.8) | 320 (99.4) | 4.3 (1.3) | 4.2 (0.8) | 321 (99.7) | 4.3 (1.3) | 4.1 (0.9) | 321 (99.7) |
| work content | 3.7 (1.0) | 3.7 (0.7) | 320 (99.4) | 3.7 (0.7) | 3.7 (0.7) | 321 (99.7) | 3.7 (1.0) | 3.7 (0.7) | 319 (99.1) |
| social relation | 4.0 (1.0) | 4.0 (0.8) | 321 (99.7) | 4.0 (1.0) | 4.0 (0.8) | 321 (99.7) | 4.0 (1.0) | 4.0 (0.8) | 321 (99.7) |

*(Continued)*

**Table 1.** (Continued)

Legend: m = mean; sd = standard deviation; IQR = interquartile range; n = absolute numbers for categorical variables or number to treat for continuous variables; % = relative frequency

[a] no included study participant with participation at all three timepoints reported diverse gender

[b] short-time work in the sense of working a reduced number of hours and receiving compensation for missing net wages

payment of wages in the case of temporary short-time work due to economic difficulties already before the COVID-19 pandemic [30,31]. In response to the pandemic, the requirements for companies in Germany to receive compensation for short-time work were eased starting in March 2020. Another relevant regulation by the German Ministry of Health was that, beginning in June 2021, occupational physicians were authorized to vaccinate in the workplace as part of the national vaccination campaign [32].

## Perceived psychosocial demands and adjusted working conditions in autumn 2020

According to the interview partners, some employees perceived existential worries at the beginning of the pandemic due to the sudden and unknown situation (IP8a, Table 2). Impacts on the economy were unclear and increased the employees' fear of job loss. To reduce the work-related risk of infection, SARS-CoV-2-infection control measures were introduced, requiring re-organization of working processes. One interview partner reported for example, that employees concerned were involved in re-organizing the assembly line to allow as-safe-as-possible workplaces while ensuring that work remained efficient (IP8b).

Survey results show that at T0, psychosocial demands from aspects of work organization were rated as moderate with 2.7 (1.0) (median (IQR)), Table 1. Particularly noticeable are professionals of the company's medical service, reporting to experience unreasonable workloads that do not match their skills and capacities (1.3 (0.8)). Fig 2a visualizes the perceived psychosocial ratings, differentiated per examined working field. Mean perceived psychosocial demands were rated more unfavorably by managers than non-managers when it came to aspects of work organization. Contrarily, managers rated aspects relating to work environment and content more favorably, and aspects of social relations equally, Fig 2b.

## Organizational structures

At the onset of the COVID-19 pandemic, an already existing crisis management team began meeting regularly to make important procedural decisions to successfully navigate the pandemic (IP6a). The team consisted of experts from various fields such as law, occupational medicine and safety, logistics, purchase, human resources, information technology, and communication. The interviews highlighted the benefits of being part of a large company group with a wide range of human resources and expertise. Interview partners also shed light on the company's communication culture. A variety of existing and new communication channels allowed every employee at every workplace to be informed about current procedures very early on after the SARS-CoV-2-outbreak (IP2a). Because of established procurement channels and financial resources within the company group, it was possible to carry out large-scale SARS-CoV-2-testing within a short time frame (IP5b).

## Changes in perceived psychosocial demands during and before the COVID-19 pandemic

Significant differences in the median perceived psychosocial demands from aspects of work organization between the three survey timepoints during the pandemic were indicated by the Friedman-Test ($X2(2)=7.04$; $p = 0.03$), see Table 1. However, the low effect size ($r = 0.01$) suggested negligible differences and post-hoc analyses revealed no significant pairwise differences. The median psychosocial demands from aspects of the work environment were perceived as rather favorable and stable ($X2(2)=5.47$; $p = 0.06$; $r = 0.09$), as were the perceived psychosocial demands in terms of work

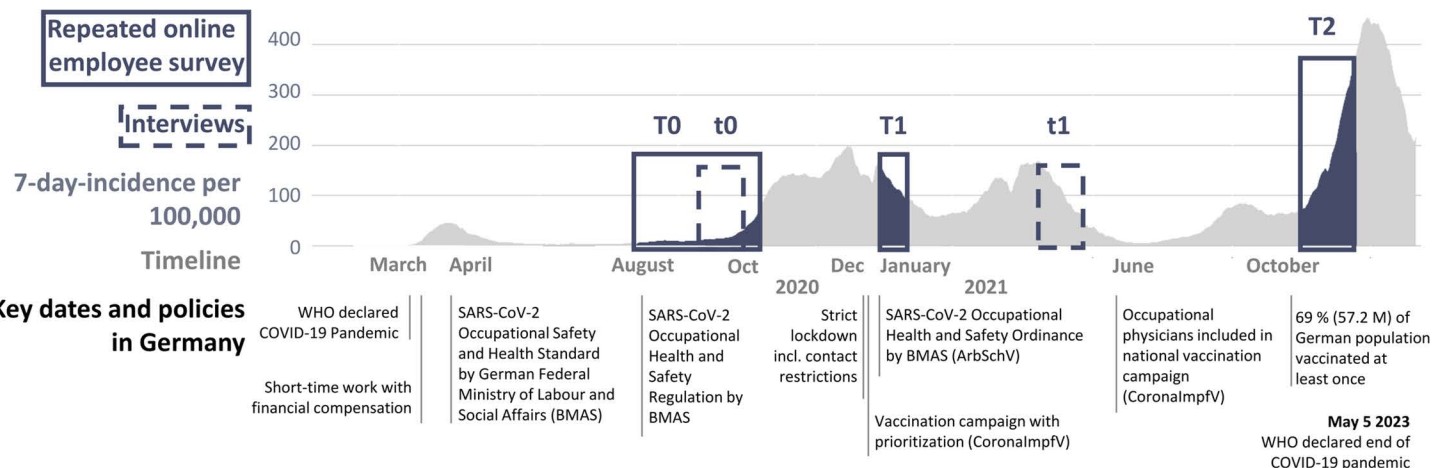

**Fig 1. COVID-19 pandemic context.** Data collection phases (T0, T1, T2: quantitative employee survey, t0 and t1: qualitative interviews with managers) and the COVID-19 pandemic context in Germany. Key dates and policies during the COVID-19 pandemic in Germany are shown with the 7-day incidence per 100,000 inhabitants.

content (X2(2)=0.45; p = 0.08; r < 0.001) and social relations (X2(2)=1.70; p = 0.43; r = 0.003). Differences between managers and non-managers regarding perceived psychosocial demands were evident and remained constant throughout the pandemic, see Fig 2b, e.g., both groups experienced an unreasonable workload during the pandemic, which was slightly more pronounced among managers. Aspects of work environment and content were rated favorably by both, slightly more pronounced by managers. Both groups reported experiencing equally supportive social relationships at work.

According to the interviews, introducing comprehensive infection control measures in the workplace involved new tasks for managers, e.g., implementing SARS-CoV-2-measures and ensuring employee adherence (IP4a). Repeated reminders to employees and face-to-face meetings to clarify expectations seemed to be essential (IP8c). Interviewed managers reported that they trusted their employees to take personal responsibility and to correctly assess health-related safety measures and organizational regulations (IP4b). They further emphasized their strategy of honestly pointing out the general lack of knowledge at that time and the need to test the applied organizational rules for effectiveness (IP4c). Furthermore, remote work was extended where possible to reduce personal contact on-site, especially for office employees. As remote work was already possible before the pandemic, capacities were expanded immediately after the outbreak of the SARS-CoV-2 (IP5c). However, as the pandemic progressed, contrary opinions on remote work evolved: some employees were in favor of the opportunity to work from home while others missed their colleagues and felt frustrated. Feelings of frustration were mainly rooted in the ineffectiveness that employees felt due to missing consultations within working groups and having little face-to-face contact (IP3a, 5a). Regarding the social relations at work, especially the high commitment by the company's medical service professionals was highlighted as well as their constant availability and support (IP7a).

Retrospectively, median perceived psychosocial demands from aspects of work organization did not significantly differ between timepoints (X2(2)=1.52; p = 0.47; r = 0.002), see Table 1. When comparing working fields, see Fig 3a, company's medical service personnel (n = 6) stood out. 67% (n = 4) perceived psychosocial demands from aspects of work organization deteriorated during the COVID-19 pandemic compared to before. Overall, about 25% of employees and managers perceived psychosocial demands from aspects of work organization during the pandemic more favorably than before, see Fig 3b. Overall, psychosocial demands from aspects of work environment were perceived favorably retrospectively and did not significantly differ between timepoints (X2(2)=12.2; p = 0.002; r = 0.02; non-significant post-hoc tests), as were

**Table 2. The organization's responses to COVID-19 pandemic-related challenges and their impact on the individual's daily work are presented in the qualitative data and the following quotations from the interview partners (IP).**

| Category<br>*Subtheme* | Quote | *Time point* |
|---|---|---|
| **Pandemic-related changes in working conditions** | | |
| *For employees* | When looking back on the year now, it's clear that the fear at the beginning of the pandemic was much greater than it is now. In the beginning, of course, the problem was that nobody knew what was going on from a medical point of view: what's happening, how serious are the consequences, what would happen if I got infected? Then, there were the issues relating to health and physical fitness. And the second thing was, of course, jobs.... well, then the economy collapsed. There really had been predictions that the economy would collapse and we would lose an enormous number of jobs. **IP8a** | t0 |
| | There are a lot of misunderstandings, you speak less, write more, get things wrong, and it builds up because there are still no discussions. These are the problems, and on the other hand, of course, there are similar cases that arise because there are fewer meetings when working remotely or less contact with colleagues when working at the company. Nothing is being coordinated, so there is a duplication of work, two or three people are working on the same thing and only realize it two days later. That's frustrating when you already have too much to do. **IP3a** | t1 |
| | […] There is also a psychological burden, employees report this as well. I'm exaggerating now, but I think you might understand that, they suffer from the fact that they have no direct contact with their colleagues. [...] On the flipside, however, many colleagues say: working from home is really good, like, I don't even know whether I want to go back. **IP5a** | t1 |
| *For managers* | For managers, I feel that their workload has increased because there is more to organize than just the day-to-day work. Things to organize like who is allowed to be on-site, which rules are to be implemented, and which additional measures need to be managed and monitored. **IP4a** | t0 |
| | […] Of course, I had individual discussions with those who are not so sensitive and I said: you have to show consideration to colleagues who are more fearful, and you have to take their concerns seriously and of course try to keep your distance and wear face-masks. **IP8c** | t0 |
| **Organizational processes to adapt working conditions** | | |
| *Decision-makers and stakeholders* | And, of course, it was very important to me to involve the employees on-site, as a team, in a workshop. I was there as the foreman, one of my shift supervisors was involved as well as five employees from the assembly lines and the [process] engineer. So, we designed it together. It was also important to me to involve the employees, especially in this situation, so that they can really voice their worries or fears and concerns. **IP8b** | t0 |
| | […] So the crisis management team basically includes various experts – there is corporate communications, [information technology] IT, [human resources] HR, the Health Safety & Environment department and the occupational safety specialists, and then whomever else might be needed. We also had labor lawyers, as well as business lawyers, because there are questions that, of course relate to a completely different – a legal – dimension, as well as the [company medical personnel] and the representatives of the [company medical personnel] team. **IP6a** | t0 |
| *Procedures and planning* | Firstly, I think there is no question that we have home office workplaces on a scale that we have never had before. We are familiar with this model. We had already introduced it, but of course not on this broad of a scale. **IP5c** | t0 |
| | […] Whenever the [new COVID-19] regulation [by the government] comes into force at short notice, like, now a rapid lateral flow test would be nice, and then all of a sudden, rapid tests are mandatory three days later, and then we try to organize that. Of course, we are part of the corporate group, which is very helpful, especially when something has to be procured on a large scale. **IP5b** | t1 |
| | As far as protection is concerned, I think we are actually very, very well positioned here. [...] So, as I said, our [company medical personnel] is always very, very committed and makes sure that we really take advantage of all opportunities. [...] And if we have any questions, like at the beginning when people called me, all the unresolved cases, saying: Yes, I also had contact with him, with the person who tested positively, or I also have slight symptoms now, what should I do? Well, I could always call the [company medical personnel] and talk about it: What do we do now? What do we do in this case? And I definitely got a lot of support there. I thought that was really great, yes, that they were always there for me to help me with any problem. **IP7a** | t1 |
| **Culture of trust** | | |
| *Self-respon-sibility* | And I explained many rules and hoped that the employees would take responsibility for themselves and this was confirmed, that they were able to manage themselves and be flexible. **IP4b** | t0 |

*(Continued)*

**Table 2.** (Continued)

| Category Subtheme | Quote | Time point |
|---|---|---|
| *Communication culture* | […] And we also started very early with communication in general, with the protective measures. This included personal letters from our management to employees, publications in our media, such as the [company newspaper], in which we regularly publish articles. And there is a weekly newsletter, one for each country and one for [the corporate group] worldwide, which we distributed. Thus, we try to share with as many employees as possible what we are working on as well as changes, including scientific findings, legal changes and so on. **IP2a** | t0 |
| | However, I also believe that it is good for communication if you say quite honestly that you don't know everything and that it is therefore possible that tomorrow you will take back or correct something that you thought was good today, so that people understand that not everything is absolute and that it changes. **IP4c** | t0 |

The interview partners covered the following perspectives: factory security service, company medical service, works council, technical operation manager, corporate financial management, human resources department, and assembly line/ manufacturing. Since we focus on the overall content, we expect no loss of meaning from the authors' translation from German into English, which was done after the analysis and checked by a native speaker. The original German wording of the items is provided in the S6 Table.

psychosocial demand ratings from aspects relating to work content (X2(2)=1.11; p = 0.58; r = 0.002) and social relations (X2(2)=3.88; p = 0.14; r = 0.006).

## Discussion

This longitudinal mixed-methods study revealed no changes in the participants' perceived psychosocial demands from aspects of work organization, work environment, work content, and social relations during the COVID-19 pandemic from autumn 2020 (COVID-19 pandemic in Germany: summer plateau and eased criteria for receiving compensation for short-time work) to January 2021 (extended remote work opportunities during a stricter lockdown), and autumn 2021 (more vaccination possibilities provided by the company's medical service). Psychosocial demands related to work organization were perceived as unfavorable, especially for managers and company's medical service personnel. In contrast, the majority of our study population reported favorable perceptions of psychosocial demands from aspects of work environment, work content, and social relations. Overall, the retrospective ratings did not differ significantly from the perspectives during the COVID-19 pandemic. Combining quantitative with qualitative results highlighted the importance of an organization's pandemic preparedness in successfully dealing with challenging situations by redesigning working conditions. Key aspects were a crisis management team, a culture of trust, extensive and transparent communication, and participatory approaches in change processes (e.g., involvement in re-organizing assembly lines).

Our results are similar to another panel study among employees not working in the healthcare sector in Germany, where reported psychosocial demands from reduced contact with colleagues and work-privacy conflicts decreased from winter and spring 2021 until winter 2022 [33]. In comparison to us, Casjens et al. [33] examined psychosocial demands as well as strain. They found that psychosocial demands such as work-privacy conflicts, negative work-related stress, and work-related SARS-CoV-2-infection risk were associated with negative strain, here anxiety- and depression-related symptoms. Reported severity of anxiety- and depression-related symptoms increased at the beginning of the COVID-19 pandemic, were highest in winter and spring 2021 but declined until winter 2022 [33]. Similarly, in a representative German study population, risk factors for increased self-reported depression and anxiety symptoms from pre-pandemic (2014–2019) to November 2020 were identified, including poorly equipped remote workplaces, high workloads resulting from having to cope with high absenteeism rates due to illness or suspected infections, younger age, work-privacy conflicts, and job insecurity [34]. Findings further show that mental strain increased less in employees who worked a reduced number of hours with compensation than employees who worked reduced hours but did not receive compensation for missing net wages [34]. From a macro-level perspective, the German social security concept included employment-preserving systems such as continued payment of wages in the case of temporary short-time work due to economic difficulties

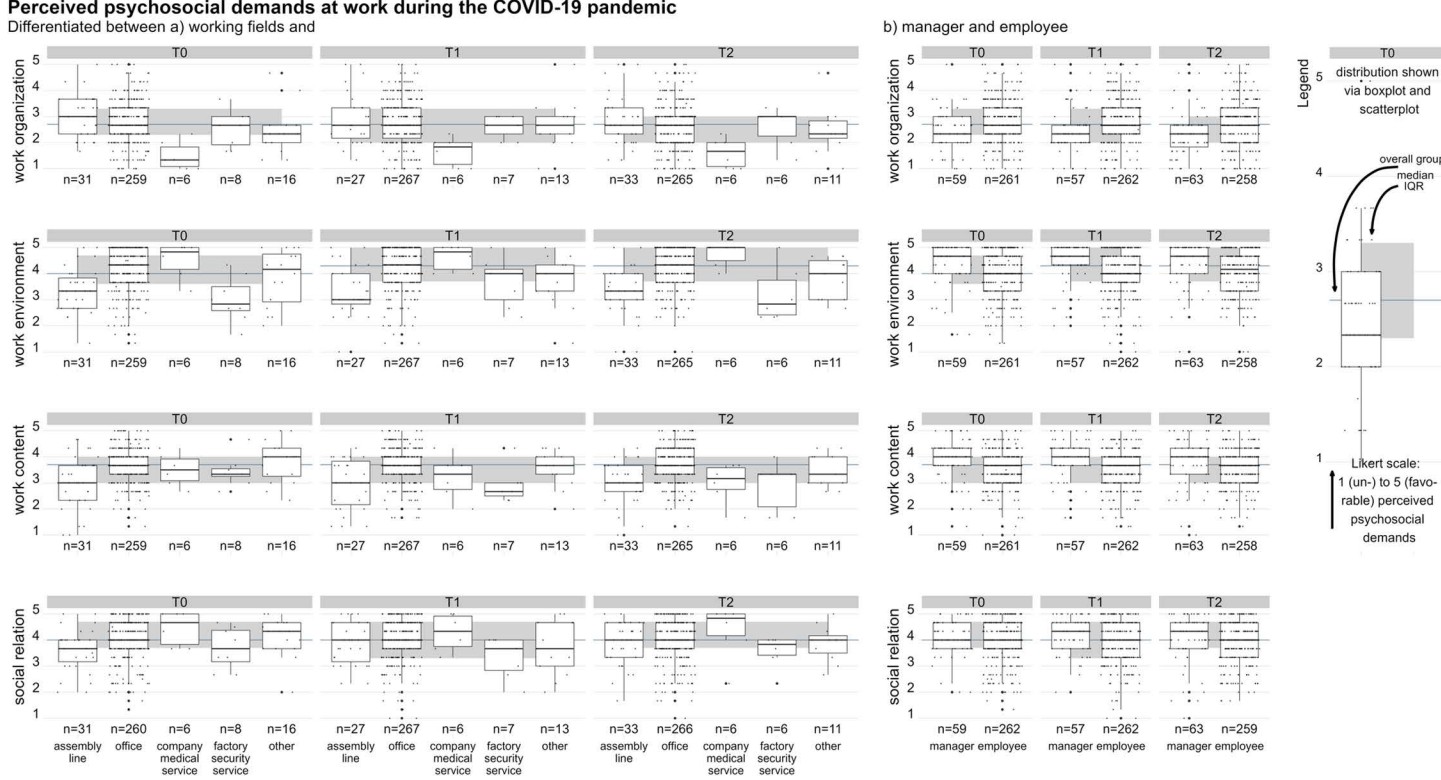

**Fig 2. Perceived psychosocial demand ratings during the COVID-19 pandemic.** *a (left)*: perceived psychosocial demand ratings for each examined working field in contrast to the overall median rating at each survey wave during the COVID-19 pandemic; *b (right)* perceived psychosocial demand ratings for managers vs. non-managers in contrast to the overall median rating at each survey wave during the COVID-19 pandemic.

already before the COVID-19 pandemic [30,31]. Access for companies in Germany to this compensation scheme were eased in March 2020 to help employers cope with pandemic-related economic difficulties. Our interviews revealed that employees across all working fields initially had existential concerns and feared job loss due to the uncertain global economic situation. Our quantitative data showed that 99.7% of all participants hold permanent work contracts. Thus, we were unable to compare their perceived psychosocial demands with employees holding fixed-term contracts. We can only hypothesize that the job security provided by the permanent work contracts across all working fields in combination with the possibility to work a reduced number of hours with compensation for missing net wage contributed to the stability of perceived work-related psychosocial demands.

## Practical implications

Our results suggest that the organization's responses to the challenges posed by the pandemic helped prevent an increase in employees' work-related psychosocial negative demands during the pandemic. Depending on the overall context of where an employee works, what worked for the organization investigated here may not work in others [35]. To better understand how working conditions can ensure work ability during challenging times, we aimed to derive practical implications for organizational pandemic preparedness from our findings (meso-level (company level)). As interactions between players exist [36], we additionally draw learnings on the macro- (societal level) and micro-level (individual level). While macro-level policies usually provide a certain framework, changes are implemented on the meso-level to ensure health-related preventive behaviors on the micro-level.

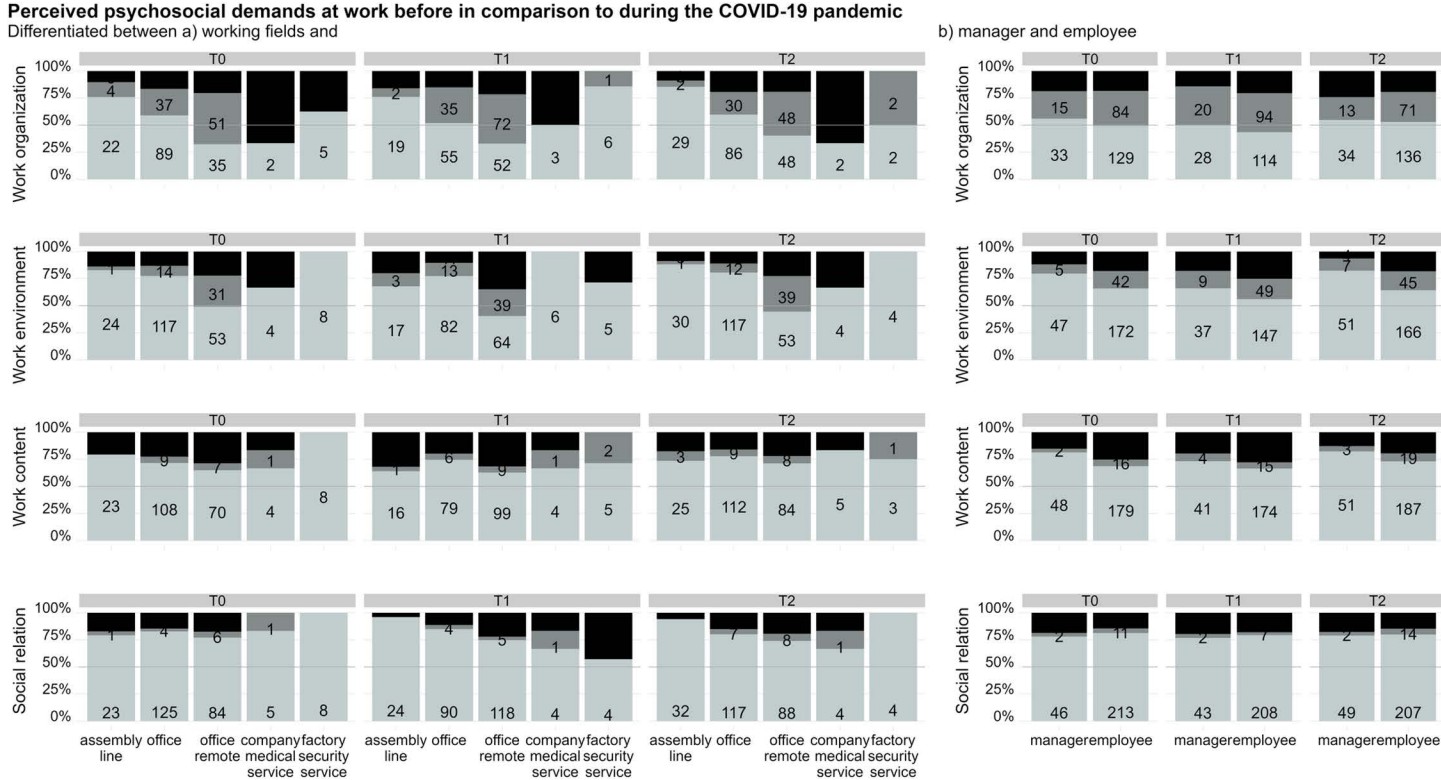

**Fig 3. Perceived psychosocial demand ratings before vs. during the COVID-19 pandemic.** *a (left)*: differences in the ratings in terms of perceived psychosocial demands before vs. during the COVID-19 pandemic differentiated per working field; *b (right)* differences in the ratings in terms of perceived psychosocial demands before vs. during the COVID-19 pandemic differentiated between manager vs. non-manager. Color coding: Light grey: during the COVID-19 pandemic perceived psychosocial demands are perceived equally to retrospectively perceived psychosocial demands at work before the COVID-19 pandemic; grey: during the COVID-19 pandemic, psychosocial demands are perceived more favorable than retrospectively perceived psychosocial demands at work before the COVID-19 pandemic; black: during the COVID-19 pandemic, psychosocial demands are perceived less favorable than retrospectively perceived psychosocial demands at work before the COVID-19 pandemic.

Starting in March 2020, recommended SARS-CoV-2-infection control measures at work were to be introduced according to the hierarchy of prevention and control (macro-level; technical and organizational take precedence over personal measures) [37,38]. In the examined organization, a crisis management team made the most important decisions (meso-level). Our interviews showed that managers struggled with increased workloads from leading measures and ensuring adherence. Implementing specific OSH-measures required one-one meetings and tailored, subgroup-specific solutions (micro-level), as there was no 'one-size-fits-all'-solution. Decisions had to be made quickly regarding which measures were to be taken in specific situations even when policies at national and state level were uncertain. As managers and employees had, on average, been employed for a long time, they were able to rely on trust and self-personal responsibility (micro-level). To maintain a long-term acceptance of adjusted working conditions, the company presented here was able to draw on many human resources, an established team of occupational health professionals, and a wide variety of communication channels (meso-level). Similar to our findings, it was previously shown in healthcare organizations, that communication was key to making employees feel safe and cared for in their workplaces [39]. Evidence suggests that the larger the organization, the more personnel, structured communication approaches, and formalized procedures are available for managing psychosocial risks at work [40,41]. This limits the generalizability of our findings. The investigated company differs from small or medium-sized companies, where limited financial capacities and lack of experts may restrict

organizational resilience [42]. Cooperation between smaller organizations or within employer associations could compensate for this. This suggestion was already stated in a pandemic plan for non-healthcare organizations provided for example by the German Social Accident Insurance in cooperation with the Federal Office for Civil Protection [43].

In December 2020, an increase in SARS-CoV-2-infections among the German population was followed by another severe lockdown in January 2021 [27]. The SARS-CoV-2 Occupational Health and Safety Ordinance [44] was passed by the Federal Ministry of Labor and Social Affairs to underscore the necessity of offering remote work wherever possible (macro-level). Again, the organization studied, benefited from structural resources as remote working possibilities had already been established and capacities could be expanded quickly (meso-level). In June 2021, occupational physicians were allowed to vaccinate in the workplace as part of the national vaccination campaign (macro- and meso-level) [32]. Our data revealed that company's medical personnel experienced increased stress from aspects of work organization since early 2020 but were highly valued by employees for their expertise and efforts (micro-level). Our findings correspond to a study evaluating pilot COVID-19-vaccination campaigns in multiple companies in Germany [45].

As the German labor system benefited from strong social security measures that were established already prior to the COVID-19 pandemic, the generalizability of our findings to other countries is limited. In countries other than Germany with weaker social security measures, it seems to be even more important for organizations to design and establish resilient work design to protect employees from severe psychosocial demands during future crises. During the COVID-19 pandemic, e.g., company medical personnel played an important role in the organizational context. This established expertise may also be critical in future public health crises.

## Strengths and limitations

Regarding strengths, this exploratory longitudinal mixed-methods study used two complementary research methods. This allowed us to comprehensively present the perspectives of different working fields, increasing the credibility of the overall findings. The panel data allowed for analyzing within-person changes between timepoints. When planning our study project in spring 2020, we considered the current state of research and included the latest findings for the COVID-19 pandemic as a new and unknown situation. A strength is that the first period of data collection was carried out during the early stage of the pandemic. As the pandemic progressed, we included a third survey wave in autumn 2021. At that time, the majority of the German population had already been vaccinated at least once. We kept the exploratory study design throughout the study project for mainly two reasons: following the codex of good scientific practice and the framework for developing and evaluating complex interventions [46]. The retrospective perspectives were assessed three times without change. Thus, we assume that subjective perceptions of work-related psychosocial demands remained stable. As the unexpected effects of the COVID-19 pandemic constitute a completely new experience, there may have been a shift in response due to a re-calibration of the person-specific internal evaluation range [47]. We cannot rule out the fact that participants might not accurately recall their perceived state before the pandemic (recall bias). We were mainly interested in how employees subjectively perceived work-related psychosocial demands. Our retrospective survey design could have led employees to rate their psychosocial demands during the pandemic as more stressful than before, e.g., due to negativity bias (focusing more on current negative experiences than on favorable ones) and 'rosy' retrospection (viewing the past more positively) [48]. Both recall and negativity bias may have contributed to an overestimation of unfavorably perceived working conditions during the pandemic compared to before, when rating the 'before' timepoint retrospectively. Indications of theses biases were apparent in the qualitative findings. For example, one interviewee reflecting on the last year (IP8a), emphasized intense negative emotions related to fear about health and economic uncertainties. This illustrates how strong intense negative emotional experiences might have shaped the retrospective evaluations. However, our results did not show any significant changes over time. In addition, the retrospective evaluations gathered in 2020 and 2021 presented here, seem to show a similar snapshot of the employees' perceived work-related well-being compared to before the pandemic. According to the internal results of a worldwide employee survey at the collaborating company group

with a response rate of 86% (n = 270.000) in 2017, the majority of participants were satisfied with their working conditions and 82% would even recommend their employer to family and friends.

Regarding limitations, we examined the perceived psychosocial demands among the employees of a large company group characterized by a well-established safety culture and large financial and human resources. The company group made it possible to produce disinfectants and protective equipment in-house at the pandemic's beginning to reduce supply shortages by equipping suppliers and own employees appropriately. Therefore, our findings may not fully be generalizable to small or medium-sized organizations with more constrained financial, technical, and human resources, as already discussed in further detail in Soeder et al. [10]. Office and assembly line work areas generally represent the largest groups of employees within a company, whereas groups such as the company medical service are naturally smaller. However, the sample size in some work areas was too small to draw statistical conclusions or generalize the results. Another limitation is the reliance on a small number of recommended dimensions for implementing a comprehensive psychosocial risk assessment at work by the GDA [7]. Results are limited in the transferability to other countries as the GDA-scheme refers to the German legislation and context. Cronbach's alpha ($\alpha = 0.5$) of the aspect of work content shows a low homogeneity of the three test items. Here, different response patterns of employees of different working fields should be considered. This aligns with the findings of Wagner et al. [49], who similarly observed different response patterns when investigating psychosocial demands among nurses compared to physicians. The low Cronbach's alpha highlights the importance of interpreting results working field-specifically, considering the work context of each. In terms of methodological limitations, the quantitative data relied on a self-reported survey; hence, common method biases may exist, including, e.g., possible sample bias.

## Future work

Following the framework for developing and adapting complex interventions [46], our findings can contribute to the development of future pandemic preparedness interventions and their implementation in occupational settings. From our point of view, the here presented findings can be transferred to other current and future global challenges requiring a high level of employee-engagement in organizations. Global challenges such as climate change, economic downturns, or topics related to demographic change have in common that they disrupt established routines and require decision-making under uncertain and dynamic conditions. In our opinion, pandemic preparedness plans not only strengthen organizational resilience during pandemic-like situations. Developed and established structures and strategies further allow faster and solution-driven reactions during other future crises. On April 17, 2025, the WHO member states finalized a draft regarding pandemic preparedness to strengthen global collaboration such as ensuring stable supply chains of medical products and to highlight the importance of developing pandemic preparedness plans on a national level [50]. This draft regarding pandemic preparedness was adopted on May, 20, 2025, and is to be ratified by all member states [51].

Demographic change and its implications for the world of work are already very prominent in Germany. Previous study findings for a representative German population sample revealed that the majority of study participants, born in 1959 or 1964, reported that they were not willing to work until retirement age [52]. The findings additionally showed that when employees perceived improved working conditions and favorable safety culture, they were more likely to delay retirement. Examined aspects of work design included, e.g., feeling appreciated for their work or a perceived personal responsibility in terms of work-time organization. Topics for future research on the topic of demographic changes could deal for example with how managers address digital competency gaps among older workers or how to support those workers' lifelong learning in rapidly changing work environments.

As previously stated [43], aspects such as extensive and transparent risk communication, employee participation via works council, management training, and personal responsibility of employees should be considered in pandemic preparedness of non-healthcare organizations. Our findings confirmed that a crisis management team, a wide variety of communication channels, a certain culture of trust, and participatory approaches in designing working conditions were

beneficial during the unprecedented situation of the COVID-19 pandemic. Thus, organizations can shape employees' experiences during a global crisis and influence their perceptions of psychosocial demands. Derived practical implications are particularly relevant for policymakers, managers, and company's medical service personnel. On an organizational level, for managers, derived practical implications highlight the importance of transparent communication, trust-based leadership styles, and participatory approaches in change processes. For company's medical service personnel, it again became clear that they played a key role in creating a certain culture of trust by contributing to decisions of the crisis management team and managing internal testing and vaccination campaigns as part of the national vaccination strategy. The hereby established culture of trust and expertise may also be helpful during future public health crises.

## Conclusions

Our findings contribute to a contextualized understanding of how shaping the design of good and sustainable working conditions can ensure work ability even during challenging times. We examined already established OSH-structures and derived practical implications that were effective for a large company group rich in resources during the COVID-19 pandemic. These can be used when developing future pandemic preparedness interventions. Based on the triangulation of qualitative and quantitative findings, we understand pandemic preparedness in psychosocial work design to be an ongoing process of deriving and implementing what has been learned to be better prepared for upcoming challenges.

## Supporting information

**S1 Table. Items used to assess the perceived psychosocial demands from aspects relating to work organization, work environment, work content, and social relations in the workplace.**
(PDF)

**S2 Table. Perceived psychosocial demands during the COVID-19 pandemic: Missing analysis of main outcome.**
(PDF)

**S3 Table. Interview guide.**
(PDF)

**S4 Table. Coding frame for the qualitative interview data.**
(PDF)

**S5 Table. Sensitivity analysis of outcome variables.**
(PDF)

**S6 Table. Quotes from qualitative interviews.**
(PDF)

## Acknowledgments

We would like to thank all the study participants for their time and effort. Furthermore, we would like to thank Lorri King, a native speaker from able Sprachschule GbR, for reviewing the grammar and wording of the manuscript. J.S. gratefully acknowledges the valuable support and feedback of her Doctoral Committee. This study is also part of the first author's (J.S.) work toward a doctoral degree.

## Author contributions

**Conceptualization:** Jana Soeder, Christine Preiser, Falko Papenfuss, Esther Rind, Monika A. Rieger.

**Data curation:** Jana Soeder, Christine Preiser.

**Formal analysis:** Jana Soeder, Christine Preiser.

**Funding acquisition:** Esther Rind, Monika A. Rieger.

**Investigation:** Jana Soeder, Christine Preiser, Anke Wagner, Anna T. Neunhöffer, Falko Papenfuss, Esther Rind, Monika A. Rieger.

**Methodology:** Jana Soeder, Christine Preiser, Anke Wagner, Anna T. Neunhöffer, Andrea Wittich, Esther Rind, Monika A. Rieger.

**Project administration:** Esther Rind, Monika A. Rieger.

**Supervision:** Esther Rind, Monika A. Rieger.

**Validation:** Jana Soeder.

**Visualization:** Jana Soeder.

**Writing – original draft:** Jana Soeder.

**Writing – review & editing:** Christine Preiser, Anke Wagner, Anna T. Neunhöffer, Falko Papenfuss, Juliane Schwille-Kiuntke, Andrea Wittich, Esther Rind, Monika A. Rieger.

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
