## [Decision Letter · Decision Letter 0]

4 Apr 2025

Dear Dr. Soeder,

Thank you for submitting your manuscript to PLOS ONE. After careful consideration, we feel that it has merit but does not fully meet PLOS ONE’s publication criteria as it currently stands. Therefore, we invite you to submit a revised version of the manuscript that addresses the points raised during the review process.

We look forward to receiving your revised manuscript.

Kind regards,

María Andrée López Gómez, Ph.D.

Academic Editor

PLOS ONE

2. Please include a copy of Table 2x which you refer to in your text on page 17.

Additional Editor Comments (if provided):

Reviewers' comments:

Reviewer's Responses to Questions

**Comments to the Author**

1. Is the manuscript technically sound, and do the data support the conclusions?

Reviewer #1: Partly

Reviewer #2: Yes

2. Has the statistical analysis been performed appropriately and rigorously?

Reviewer #1: Yes

Reviewer #2: Yes

3. Have the authors made all data underlying the findings in their manuscript fully available?

Reviewer #1: Yes

Reviewer #2: Yes

4. Is the manuscript presented in an intelligible fashion and written in standard English?

Reviewer #1: Yes

Reviewer #2: Yes

Reviewer #1: Line 69-70: Explanation for psychosocial related work risks needed

Line 289-291: Concerns around small sample size; power analysis/controlled variability/effect size should be shared to address inclusion of small Ns

The paper either needs more justification for the inclusion/choice of various working fields or to focus on a specific field. Unclear from the paper how the different fields would experience comparable psychosocial demands, especially when some quotes reflect unique work environments. Small Ns in statistical analysis beg the question as to why you included the various fields rather than focusing on office workers for the paper. That these are a part of one organization should be explained earlier on--with a description of the company.

Additionally, more clarity on German supports for workers during the time could be helpful and should be addressed in statistical analysis. If supports vary by working field, this should also be addressed.

Reviewer #2: Introduction

Strengths:

Contextual Relevance:

The inclusion of current global drivers like digitalization, ecological transformation, and demographic changes is highly relevant. The connection to how these factors impacts work environments and psychosocial risks is timely and significant.

Clear Problem Statement:

The problem you aim to address is clearly articulated: the relationship between organizational responses to the pandemic and employees' perceived psychosocial demands. This is an important and under-explored area in occupational health and safety research.

Comprehensive Literature Review:

You effectively refer to prior research, including the work of the ILO, OECD, and previous crises like the Ebola epidemic and the 2008 economic recession, which supports the urgency of understanding psychosocial risks in the workplace. The inclusion of both physical and psychosocial demands aligns well with the focus of your paper.

Use of Case Examples:

Your examples, such as the changes in working conditions during the COVID-19 pandemic (e.g., remote work, social distancing), help to ground the theoretical framework in real-world changes. This provides practical relevance and context to your research.

Methodological Approach:

The combination of qualitative and quantitative methods adds depth to your investigation. It seems like you're trying to bridge a knowledge gap by exploring both employee and manager perceptions during the pandemic.

Suggestions for Improvement:

Clarity and Flow:

The first few sentences could benefit from more clarity. For example, the initial reference to ILO and OECD could be better connected to the rest of the paragraph to avoid abruptness.

"According to the International Labor Organization (ILO) and the Organization for Economic Co-operation and Development (OECD), increasing digitalization, ecological transformation, and advancing demographic change are key drivers of accelerated technological development and the constantly changing world of work."

This sentence could be shortened and made more direct for easier readability, such as:

"The International Labor Organization (ILO) and the Organization for Economic Co-operation and Development (OECD) highlight that digitalization, ecological transformation, and demographic changes are reshaping the world of work."

Language and Grammar:

The phrasing in certain areas could be streamlined. For example:

“Psychosocial risk factors in the workplace such as long working hours or employment uncertainties have been shown to be strongly associated with the development of health issues and chronic diseases, e.g., burnout, coronary heart diseases, and anxiety.”

This could be rephrased as:

“Psychosocial risk factors, including long working hours and employment uncertainty, are strongly associated with health issues such as burnout, coronary heart disease, and anxiety.”

Engagement with Data:

You refer to survey findings and telephone interviews but do not provide much detail on the data collection process. Briefly mentioning the survey method (e.g., sample size, how it was administered) could help the reader better understand the research design.

Link Between Concepts:

The transition from organizational responses to employee health could be more seamless. Right now, it jumps from one topic to another without much elaboration. It may be helpful to add a sentence or two linking how the organizational responses directly influence employee perceptions of psychosocial demands.

MATERIAL AND METHODS

You mention using both qualitative and quantitative methods, but this section would benefit from a clearer outline of how each method complements the other. For example, you could describe briefly how the qualitative data (e.g., interviews) and quantitative survey data will be integrated in your analysis.

Clarity in Structure and Flow:

The first two sentences are somewhat dense and can be broken up for better readability.

“This study is part of a larger explorative modular mixed-methods study project examining the acceptance of occupational SARS-CoV-2-infection control measures (ICM) (blinded).”

This could be more direct: “This study is part of a larger, explorative, modular mixed-methods research project that investigates the acceptance of SARS-CoV-2 infection control measures (ICM) in occupational settings (blinded).”

Clarify "Blinded" Information:

You mention "blinded" but do not specify what is blinded. Is it the participant group, the intervention, or something else? It would be clearer if this were explicitly stated.

Provide More Detail on the Mixed-Methods Approach:

The description of the mixed-methods approach can be more detailed. For example:

How were the quantitative surveys designed (e.g., Likert scales, multiple-choice questions)?

What was the interview process for the qualitative data? Were there semi-structured interviews, and how were these analyzed (e.g., thematic analysis)?

Providing more insight into these aspects would help the reader understand the depth of your approach.

Sampling Details:

You mention that participants come from various working fields but do not specify how they were selected.

“Employees and managers from various working fields...” could benefit from clarification on the sampling method (e.g., purposive sampling, random sampling). Were these employees chosen based on specific criteria?

Additionally, it would be useful to include the sample size for both the employee survey and the interviews with managers.

Participant Inclusion Criteria:

The inclusion criteria (e.g., minimum age of 18, B1-level German language proficiency, and informed consent) are clear, but consider adding any exclusion criteria (e.g., those with severe language barriers or any other reasons for exclusion).

Consider rephrasing for clarity:

“All participants were at least 18 years old, had a German language proficiency of at least B1, and provided written informed consent.”

Ethical Considerations:

The mention of ethical approval is good, but consider providing more detail on the ethical guidelines followed, such as whether any anonymization or confidentiality measures were implemented.

Result & Discussion

Clarity and Coherence:

The discussion effectively presents the findings but could benefit from clearer transitions between different sections. For instance, the shift from describing study results to comparing them with other research could be smoother. Consider using transition phrases like "In line with previous findings..." or "Consistent with prior studies, our results suggest..."

Depth of Analysis:

While the study highlights key psychosocial demands, it would be beneficial to provide a more nuanced interpretation of why no significant changes were observed over time. Were there specific coping mechanisms, employer interventions, or cultural factors that may have contributed to this stability?

Comparisons with Existing Literature:

The discussion compares findings with other studies, which strengthens its validity. However, some comparisons (e.g., with anxiety and depression symptom trends) seem somewhat broad. It would be helpful to specify how these studies align with your research focus on work-related psychosocial demands.

Contextualization of Findings:

The section on German social security measures is relevant but could be more directly linked to your findings. Did employees who feared job loss despite holding permanent contracts also exhibit higher psychosocial stress? If so, how does this inform policy or workplace interventions?

Limitations and Future Research Directions:

The discussion does not explicitly address the limitations of the study. Consider adding a paragraph acknowledging potential limitations (e.g., sample size, generalizability, or reliance on self-reported data) and suggesting areas for future research.

Practical Implications:

The findings suggest key organizational strategies (e.g., crisis management teams, participatory approaches). It would be useful to elaborate on how these strategies can be applied beyond the COVID-19 context to improve workplace well-being in future crises.

Practical Implications

Clarification of Multi-Level Approach:

The section discusses macro-, meso-, and micro-level responses, but the distinction between these levels is sometimes unclear. Consider adding a brief introductory sentence explaining how each level interacts and contributes to overall organizational resilience.

For example: "Our findings highlight the importance of a multi-level response to crisis management, where macro-level policies provide the framework, meso-level decisions implement changes, and micro-level adaptations ensure individualized solutions."

Stronger Link to Broader Workplace Implications:

The discussion primarily focuses on the specific organization studied. To make the findings more generalizable, explicitly address how these lessons can apply to organizations of varying sizes, industries, and resource levels.

Comparing to International Contexts:

The practical implications are framed within the German labor system, which benefits from strong social security measures. Briefly acknowledge how organizations in countries with weaker labor protections might face different challenges and require alternative strategies.

Strengths and Limitations

Addressing Possible Biases More Explicitly:

The discussion acknowledges recall bias and negativity bias but could be more explicit in explaining how these biases might have influenced results. For example, were there any trends in the qualitative data that suggest participants’ perceptions changed over time?

Also, the discussion of Cronbach’s alpha (α=0.5) for the work content aspect suggests low reliability. How does this impact the interpretation of results?

Limitations on Generalizability:

The study is based on a single company with pre-existing structural resources. This should be explicitly mentioned as a limitation in terms of external validity, particularly for small and medium-sized enterprises (SMEs).

Consider adding: "The findings may not fully generalize to SMEs or organizations in different sectors where financial and human resources are more constrained."

Future Work

More Specific Recommendations for Future Studies:

The section discusses potential extensions of this research, particularly in relation to demographic change. However, it would be helpful to outline specific research questions or methodologies that future studies could use.

Expanding the Scope Beyond Pandemics:

The discussion touches on other global challenges (e.g., demographic changes). It would be helpful to explicitly state how the findings might inform crisis management for challenges like climate change, economic downturns, or geopolitical instability.

Conclusions

Clearer Takeaways for Different Stakeholders:

The conclusion emphasizes organizational preparedness but could be more actionable by addressing specific stakeholders. Consider framing the key takeaways separately for policymakers, managers, and employees.

**Do you want your identity to be public for this peer review?** For information about this choice, including consent withdrawal, please see our Privacy Policy

Reviewer #1: No

Reviewer #2: No

---

## [Author Response · Author response to Decision Letter 1]

23 Apr 2025

Dear María Andrée López Gómez,

many thanks for the valuable review process and for the opportunity to submit our revised manuscript.

Please find below our answers to the reviewers’ comments on our manuscript titled ‘Pandemic preparedness in shaping psychosocial working conditions – insights for occupational safety and health from a longitudinal mixed-methods study during the COVID-19 pandemic’.

In addition, we uploaded a marked-up copy of our manuscript to highlight the changes made, labeled 'Revised Manuscript with Track Changes' as well as an unmarked version of the revised manuscript labeled ‘Manuscript’.

We checked that our manuscript meets PLOS ONE’s style requirements as well as our reference list.

We believe that we have replied to all of the reviewers' valuable comments and suggestions. Thank you very much for your kind consideration and we look forward to hearing from you.

Kind regards on behalf of all authors,

Jana Soeder (Corresponding author)

Research Associate and PhD Candidate Experimental Medicine

Institute of Occupational and Social Medicine and Health Services Research

University Hospital Tübingen

Wilhelmstrasse 27

72074 Tuebingen, Germany

Dear Reviewer 1,

thank you your time and effort in evaluating our manuscript. We have revised the manuscript based on your comments with tracked changes to highlight the revisions. The lines indicated refer to the new lines in the revised manuscript. Please find below a point-by-point response to your comments and suggestions.

Comment 1

Line 69-70: Explanation for psychosocial related work risks needed

Answer 1

Many thanks for your recommendation. We have revised the related paragraph in the introduction to provide a clearer understanding

Changes in the manuscript

Introduction: […] In Germany, OSH is regulated by the Safety and Health at Work Act (ArbSchG) [6]. Here, a risk assessment is the first step in deriving and implementing OSH-measures aimed at preventing work-related accidents and health hazards. Possible work-related risks can be elicited by both, physical and psychosocial demands. Each aspect of psychosocial demands can – depending on the individual’s preconditions – lead to either favorable (e.g., activating state) or unfavorable strain (e.g., increased work-related stressors) and thus impact the individual’s health [7]. Psychosocial demands in the workplace refer to the key aspects of work organization, work environment, work content, social relations, and new forms of work. These aspects are to be considered when When conducting a psychosocial risk assessment in the workplace, according to the the Joint German OSH-Strategy (GDA) recommends considering key aspects of work organization, work environment, work content, social relations, and new forms of work [7]. […] (line 67 ff.) 

Comment 2

Line 289-291: Concerns around small sample size; power analysis/controlled variability/effect size should be shared to address inclusion of small Ns

The paper either needs more justification for the inclusion/choice of various working fields or to focus on a specific field. Unclear from the paper how the different fields would experience comparable psychosocial demands, especially when some quotes reflect unique work environments. Small Ns in statistical analysis beg the question as to why you included the various fields rather than focusing on office workers for the paper. That these are a part of one organization should be explained earlier on--with a description of the company.

Additionally, more clarity on German supports for workers during the time could be helpful and should be addressed in statistical analysis. If supports vary by working field, this should also be addressed.

Answer 2

Many thanks for that comment. We can see your point and agree. We now added the effect size not only for statistically significant results but also for non-significant results, see lines 339-341 and lines 367-377. However, we would like to highlight that until now a lack of longitudinal panel studies investigating psychosocial demands during the COVID-19 pandemic exists. Especially the working field of non-healthcare organizations is under-explored as well as the differentiation between employees and managers of different occupational working fields.

We added a paragraph describing the company group. In addition, we included a paragraph to justify the inclusion of various working fields. We recruited from six different company sites which allows to differentiate and adhere to pseudonymization guidelines, even though the sample size in some working fields is small. However, it is important to consider that office and assembly line working fields typically represent the largest employee groups within a company, whereas fields such as company medical services are naturally smaller by comparison. As previous literature by Wagner et al. (doi: 10.3390/vaccines11061082) showed that occupational health personal suffered form increased workload especially during the roll-out phase of the COVID-19 workplace vaccination program in Germany, it seemed of utmost importance for us to include workers related to this working field in our sample if possible.

In the overall survey, N=5,554 employees participated. For 322 out of them, longitudinal panel data could be ensured. See section Results sample characteristics for further details. We added an additional sensitivity analysis to show that the outcome variables did not differ substantially between participants who took part three times compared to the overall sample, see S5 Tab in the Supplementary Material.

Changes in the manuscript

Methods section: The longitudinal data comes from employees and managers from various working fields from a worldwide leading global supplier of technology and services. The company group can be characterized as financially well-established and having a lot of human resources. Regarding the work environment, digitalized solutions and proven communication channels were already used before the COVID-19 pandemic. Employees work in various working fields (e.g., assembly line/production, office, company medical service, factory security service) of the same company group. We aim to differentiate between those occupational groups as they show large differences in how working conditions were adapted to ensure as-safe-as-possible workplaces. While office workers were mainly affected by remote work and online meetings where possible, assembly line/manufacturing workers experienced redesigned working conditions to allow to keep a minimum of 1.5m distance between work stations, frequent cleaning of work equipment, and decoupled break times. Company medical service personal in comparison were responsible for implementing SARS-CoV-2-infection control measures as well as ensuring testing and vaccination possibilities. (l. 146 ff.)

Dear Reviewer 2,

thank you very much for your valuable and comprehensive feedback on our manuscript which we appreciate a lot. We have reworked the manuscript and hope that the adaptations will meet your criticism. We revised the manuscript with tracked changes to highlight the revisions. The lines indicated refer to the new lines in the revised manuscript. Please find below a point-by-point response to your comments and suggestions.

Comment 1

Introduction - Strengths:

Contextual Relevance: The inclusion of current global drivers like digitalization, ecological transformation, and demographic changes is highly relevant. The connection to how these factors impacts work environments and psychosocial risks is timely and significant.

Clear Problem Statement: The problem you aim to address is clearly articulated: the relationship between organizational responses to the pandemic and employees' perceived psychosocial demands. This is an important and under-explored area in occupational health and safety research.

Comprehensive Literature Review: You effectively refer to prior research, including the work of the ILO, OECD, and previous crises like the Ebola epidemic and the 2008 economic recession, which supports the urgency of understanding psychosocial risks in the workplace. The inclusion of both physical and psychosocial demands aligns well with the focus of your paper.

Use of Case Examples: Your examples, such as the changes in working conditions during the COVID-19 pandemic (e.g., remote work, social distancing), help to ground the theoretical framework in real-world changes. This provides practical relevance and context to your research.

Methodological Approach: The combination of qualitative and quantitative methods adds depth to your investigation. It seems like you're trying to bridge a knowledge gap by exploring both employee and manager perceptions during the pandemic.

Suggestions for Improvement:

Clarity and Flow: The first few sentences could benefit from more clarity. For example, the initial reference to ILO and OECD could be better connected to the rest of the paragraph to avoid abruptness.

"According to the International Labor Organization (ILO) and the Organization for Economic Co-operation and Development (OECD), increasing digitalization, ecological transformation, and advancing demographic change are key drivers of accelerated technological development and the constantly changing world of work."

This sentence could be shortened and made more direct for easier readability, such as:

"The International Labor Organization (ILO) and the Organization for Economic Co-operation and Development (OECD) highlight that digitalization, ecological transformation, and demographic changes are reshaping the world of work."

Answer 1

Thank you for highlighting the importance of the research topic we addressed in our study. Thank you for this helpful remark. We revised the first sentence of the introduction to ensure a better understanding and easier readability.

Changes in the manuscript

Introduction: The International Labor Organization (ILO) and Organization for Economic Co-operation and Development (OECD) highlight that digitalization, ecological transformation, and advancing demographic change are reshaping the world of work […] (line 52 ff.).

Comment 2

Language and Grammar: The phrasing in certain areas could be streamlined. For example:

“Psychosocial risk factors in the workplace such as long working hours or employment uncertainties have been shown to be strongly associated with the development of health issues and chronic diseases, e.g., burnout, coronary heart diseases, and anxiety.”

This could be rephrased as: “Psychosocial risk factors, including long working hours and employment uncertainty, are strongly associated with health issues such as burnout, coronary heart disease, and anxiety.”

Answer 2

We revised the mentioned sentence. In addition, we went through the whole manuscript to improve the readability by shortening and rephrasing some sentences. Furthermore, a native speaker reviewed grammar and wording of the manuscript.

Changes in the manuscript

Introduction: […] Psychosocial risk factors, including long working hours or and employment uncertainty, are strongly associated with health issues such as burnout, coronary heart diseases, and anxiety […] (line 59 ff.)

Comment 3

Engagement with Data: You refer to survey findings and telephone interviews but do not provide much detail on the data collection process. Briefly mentioning the survey method (e.g., sample size, how it was administered) could help the reader better understand the research design.

Answer 3

Many thanks for that comment. We can see your point and agree. However, as part of the overall study project (see methods section ‘study design’) a study protocol, one paper explaining in detail the quantitative methods including survey design and recruiting process as well as one related results paper have already been published. These are cited as “(blinded reference)” throughout the manuscript but have been temporarily removed for the double-blind review process. To enhance clarity, we now added relevant background information to the present mixed-methods manuscript. For more details, we refer to previously published work.

Changes in the manuscript

Methods section ‘Quantitative data collection and analysis’: […] The questionnaire was developed via the online survey tool Unipark provided by Questback AS [15]. In collaboration with the company’s corporate communications department, we distributed survey invitations to all employees via email, newsletter, intranet, postcards, and posters containing a link and QR code. Participation was voluntary and we used convenience sampling. Multiple observations were linked to the same individual using self-generated anonymized codes. Details on recruiting strategies, surveyed items, and response rate at T0 (22%) are described elsewhere (blinded reference). […] (line 178 ff.) 

Comment 4

Link Between Concepts: The transition from organizational responses to employee health could be more seamless. Right now, it jumps from one topic to another without much elaboration. It may be helpful to add a sentence or two linking how the organizational responses directly influence employee perceptions of psychosocial demands.

Answer 4

Many thanks. We now hope to have clarified how the COVID-19 pandemic and related infection control measures lead to changes in working conditions and associated psychosocial demands. Furthermore, we aimed to show how organizations can influence the employees’ experiences and help buffer negative psychosocial demands throught the dynamic course of the pandemic.

Changes in the manuscript

Introduction: […] Thus, daily working conditions changed rapidly as a result of the implemented SARS-CoV-2-infection control measures. Increased remote work, physical distancing on-site, and increased cleaning and hygiene protocols led to difficulties in communication flows, blurred boundaries between work and private life, unclear responsibilities, economic and employment instability, and a dynamic pandemic situation requiring ongoing adaptations in OSH-measures. Those changes in how and under which conditions people work, impacted the work-related psychosocial demands employees experienced during the pandemic. […] (line 91 ff.)

Introduction: […] Organizational responses include, for example, communication strategies, approaches to re-designing on-site work environments, and the adaptation of infection control measures tailored to specific occupational groups. […] (line 114 ff.)

Comment 5

MATERIAL AND METHODS - You mention using both qualitative and quantitative methods, but this section would benefit from a clearer outline of how each method complements the other. For example, you could describe briefly how the qualitative data (e.g., interviews) and quantitative survey data will be integrated in your analysis.

Answer 5

Thank you very much for this helpful remark. We revised the methods section and provided more details on how quantitative and qualitative data were integrated. In addition, we highlighted the main advantage of the triangulation for our manuscript.

Changes in the manuscript

Methods section ‘Qualitative data collection and analysis’: Data integration was realized in the sampling strategy, as both, quantitative and qualitative data, were collected via convenience sampling from the same occupational groups to ensure comparability. Further, integration was realized during the exploratory analysis and interpretation phase. The triangulation of methods allowed to investigate consistent patterns in perceived psychosocial demands within occupational groups as well as between employees and managers. Qualitative data provided additional contextual insights for a more nuanced understanding of the patterns observed in the quantitative data. (line 138 ff.) 

Comment 6

Clarity in Structure and Flow: The first two sentences are somewhat dense and can be broken up for better readability.

“This study is part of a larger explorative modular mixed-methods study project examining the acceptance of occupational SARS-CoV-2-infection control measures (ICM) (blinded).”

This could be more direct: “This study is part of a larger, explorative, modular mixed-methods research project that investigates the acceptance of SARS-CoV-2 infection cont

---

## [Editor Report · Decision Letter 1]

30 Apr 2025

Dear Dr. Soeder,

Thank you for submitting your manuscript to PLOS ONE. After careful consideration, we feel that it has merit but does not fully meet PLOS ONE’s publication criteria as it currently stands. Therefore, we invite you to submit a revised version of the manuscript that addresses the points raised during the review process.

Your manuscript was thoroughly reviewed by two researchers and we would appreciate that you address the comments from each reviewer. We also consider it important to include in the title of your manuscript the region where the research was done. The title as it stands makes generalizations about psychosocial conditions overall. It is not clear in the methods section where the research was done, in one company or several in Germany. This is not a global study including various countries, hence the name of the place (in this case Germany) needs to be included in the title. If the research was done in one organization then this is also important to clarify in the title and/or abstract. Please read the following paper for recommendations: https://www.pnas.org/doi/full/10.1073/pnas.2119373119

We look forward to receiving your revised manuscript.

Kind regards,

María Andrée López Gómez, Ph.D.

Academic Editor

PLOS ONE
---

## [Author Response · Author response to Decision Letter 2]

6 May 2025

Dear María Andrée López Gómez,

many thanks for the valuable review process and for the opportunity to submit our revised manuscript.

Please find below our answers to the reviewers’ comments on our manuscript now titled ‘Pandemic preparedness in shaping psychosocial working conditions – insights for occupational safety and health from a longitudinal mixed-methods study during the COVID-19 pandemic at six company sites of one organization in Germany’.

In addition, we uploaded a marked-up copy of our manuscript to highlight the changes made, labeled 'Revised Manuscript with Track Changes' as well as an unmarked version of the revised manuscript labeled ‘Manuscript’.

We checked that our manuscript meets PLOS ONE’s style requirements as well as our reference list. As we identified further relevant literature which was published since we submitted this manuscript, we added one source to the reference list.

We believe that we have replied to all of the reviewers' valuable comments and suggestions. I hope to have now clarified in the abstract and the methods section where we conducted our research and that employees and managers were employed at six German company sites in the federal states of Bavaria, Baden-Wurttemberg, and Lower Saxony of one larger company group. Please see pages 2-5 for our reponses to Reviewer 1 and pages 6-28 for our responses to Reviewer 2. Thank you very much for your kind consideration and we look forward to hearing from you.

Kind regards on behalf of all authors,

Jana Soeder (Corresponding author)

Research Associate and PhD Candidate Experimental Medicine

Institute of Occupational and Social Medicine and Health Services Research

University Hospital Tübingen

Wilhelmstrasse 27

72074 Tuebingen, Germany

Dear Reviewer 1,

thank you for your time and effort in evaluating our manuscript. We have revised the manuscript based on your comments with tracked changes to highlight the revisions. The lines indicated refer to the new lines in the revised manuscript. Please find below a point-by-point response to your comments and suggestions.

Comment 1

Line 69-70: Explanation for psychosocial related work risks needed

Answer 1

Many thanks for your recommendation. We have revised the related paragraph in the introduction to provide a clearer understanding.

Changes in the manuscript

Introduction: […] In Germany, OSH is regulated by the Safety and Health at Work Act (ArbSchG) [6]. Here, a risk assessment is the first step in deriving and implementing OSH-measures aimed at preventing work-related accidents and health hazards. Possible work-related risks can be elicited by both, physical and psychosocial demands. Psychosocial demands can – depending on the individual’s preconditions – lead to either favorable (e.g., activating state) or unfavorable strain (e.g., fatigue or problems concentrating) and thus impact the individual’s health [7]. Psychosocial demands in the workplace refer to key aspects of work organization, work environment, work content, social relations, and new forms of work. An aspect of work organization, for example, is the constant interruption during work, while an aspect of work content is the degree of influence a worker has over assigned tasks. These aspects are to be considered when conducting a psychosocial risk assessment in the workplace, according to the Joint German OSH-Strategy (GDA) [7]. […] (lines 70 ff.) 

Comment 2

Line 289-291: Concerns around small sample size; power analysis/controlled variability/effect size should be shared to address inclusion of small Ns

The paper either needs more justification for the inclusion/choice of various working fields or to focus on a specific field. Unclear from the paper how the different fields would experience comparable psychosocial demands, especially when some quotes reflect unique work environments. Small Ns in statistical analysis beg the question as to why you included the various fields rather than focusing on office workers for the paper. That these are a part of one organization should be explained earlier on--with a description of the company.

Additionally, more clarity on German supports for workers during the time could be helpful and should be addressed in statistical analysis. If supports vary by working field, this should also be addressed.

Answer 2

Many thanks for that comment. We can see your point and agree. We now added the effect size not only for statistically significant results but also for non-significant results, see lines 371-377 and lines 403-414. However, we would like to highlight that until now a lack of longitudinal panel studies investigating psychosocial demands during the COVID-19 pandemic exists. Especially the working field of non-healthcare organizations is under-explored as well as the differentiation between employees and managers of different occupational working fields.

We added a paragraph describing the company group, see lines 160-170. In addition, we included a paragraph to justify the inclusion of various working fields, see lines 170-178. Previous findings from the European Working Conditions Survey revealed differences between working fields with respect to adjusted working conditions and the challenges associated with them (doi: 10.2806/056613). We here went into more detail for description in lines 112 ff. We recruited from six different company sites of one larger company group which allow to differentiate and adhere to pseudonymization guidelines, even though the sample size in some working fields is small. However, it is important to consider that office and assembly line working fields typically represent the largest employee groups within a company, whereas fields such as company medical services are naturally smaller by comparison. As previous literature by Wagner et al. (doi: 10.3390/vaccines11061082) showed that occupational health personnel suffered from increased workload, especially during the roll-out phase of the COVID-19 workplace vaccination program in Germany, it seemed of utmost importance for us to include workers related to this working field in our sample if possible.

We hope to have now made clear within the title, abstract and methods section, that all employees were recruited from six different company sites of one larger company group located in Germany.

Regarding the sample size, in the overall survey, N=5,554 employees participated. For 322 out of them, longitudinal panel data could be ensured. See section ‘Results sample characteristics’ for further details, lines 274 ff.. We added an additional sensitivity analysis to show that the outcome variables did not differ substantially between participants who took part three times compared to the overall sample, see S5 Tab in the Supplementary Material. Regarding the qualitative interviews, we considered a sample of nine interview participants sufficient to reach data saturation.

Regarding the support of both the German state and the employer, we on the one hand performed a comprehensive document analysis of prevailing national and federal laws and OSH-regulations for infection control in Germany. On the other hand, the interviews helped to address how workers experienced support from both the German state (e.g., financial aid in the form of paid temporary short-time work) and their organization (e.g., adjusted work arrangements) during the pandemic. Both parts provided the contextual background for interpreting and discussing our findings. We now tried to make this more clear when describing the study design, see lines 141 ff.. In addition, we went more into detail in the results section, especially when describing the main findings of the document analysis, see lines 300 ff. Less than 7% of the participants included reported working short-time, which limited our ability to analyze differences across working fields on that topic. However, we did provide an in-depth description how infection control measures were implemented across different working fields. These findings were also more relevant to the aim of our study.

Changes in the manuscript

Results section: The median psychosocial demands from aspects of the work environment were perceived as rather favorable and stable (X2(2)=5.47; p=0.06; r=0.09), as were the perceived psychosocial demands in terms of work content (X2(2)=0.45; p=0.08; r<0.001) and social relations (X2(2)=1.70; p=0.43; r=0.003). (lines 374-377)

Results section: Overall, psychosocial demands from aspects of work environment were perceived favorably retrospectively and did not significantly differ between timepoints (X2(2)=12.2; p=0.002; r=0.02; non-significant post-hoc tests), as were psychosocial demand ratings from aspects relating to work content (X2(2)=1.11; p=0.58; r=0.002) and social relations (X2(2)=3.88; p=0.14; r=0.006). (lines 410-414)

Methods section: The longitudinal data comes from employees and managers from various working fields from a worldwide leading global supplier of technology and services. The company group can be characterized as financially well-established and having a lot of human resources. Regarding the work environment, digitalized solutions and proven communication channels were already used before the COVID-19 pandemic. All employees and managers included were employed at six German company sites in the federal states of Bavaria, Baden-Wurttemberg, and Lower Saxony. Employees for example worked at the assembly line/production, in the office, company medical service, or factory security service of the same company group. We aim to differentiate between those working fields as they show large differences in how working conditions were adapted to ensure as-safe-as-possible workplaces. While office workers were mainly affected by remote work and online meetings where possible, assembly line/manufacturing workers experienced redesigned working conditions to allow for keeping a minimum of 1.5m distance between work stations, frequent cleaning of work equipment, and decoupled break times. Company medical service personnel in comparison were responsible for implementing SARS-CoV-2-infection control measures as well as ensuring testing and vaccination possibilities. (l. 160 ff.)

Title: Pandemic preparedness in shaping psychosocial working conditions – insights for occupational safety and health from a longitudinal mixed-methods study during the COVID-19 pandemic at six company sites of one organization in Germany (l. 1 ff.)

Results section: The here presented survey results are based on a subsample of participants (n=322 out of N=5,554) for whom all three survey observations were available. A total of 5,554 employees participated in the online survey at one, two, or all three timepoints. For 322 out of them, longitudinal panel data could be ensured. This analysis focuses on panel data, as we were primarily interested in within-person changes in psychosocial demands.

Methods section: For contextualization, we performed a comprehensive document analysis of prevailing national and federal laws and OSH-regulations for infection control in Germany. In a parallel approach, quantitative (employee survey) and qualitative data (interviews with managers) were gathered and analyzed independently. This allowed us to gain detailed insights into the company-level context in which employees and managers work. Findings from the interviews further helped to address how workers of different working fields experienced support from both the German state (e.g., financial aid in the form of paid temporary short-time work) and their organization (e.g., adjusted work arrangements) during the pandemic. (lines 138 ff.)

Results section: Fig 1 provides an overview of the main findings of the document analysis together with the data collection phases [22-24]. For example. the German social security concept included statutory employment-protection systems such as continued payment of wages in the case of temporary short-time work due to economic difficulties already before the COVID-19 pandemic [25, 26]. In response to the pandemic, the requirements for companies in Germany to receive compensation for short-time work were eased starting in March 2020. Another relevant regulation by the German Ministry for Health is that, beginning in June 2021, occupational physicians were authorized to vaccinate in the workplace as part of the national vaccination campaign [27]. (lines 300-308)

Discussion: Our quantitative data showed that 99.7% of all participants hold permanent work contracts. Thus, we were unable to compare their perceived psychosocial demands with employees holding fixed-term contracts. We can only hypothesize that the job security provided by the permanent work contracts across all working fields in combination with the possibility to work a reduced number of hours with compensation for missing net wage contributed to the stability of perceived work-related psychosocial demands. (l. 473 ff.)

Methods: While office workers were mainly affected by remote work and online meetings where possible, assembly line/manufacturing workers experienced redesigned working conditions to allow for keeping a minimum of 1.5m distance between work stations, frequent cleaning of work equipment, and decoupled break times. Company medical service personnel in comparison were responsible for implementing SARS-CoV-2-infection control measures as well as ensuring testing and vaccination possibilities. (lines 172 ff.) 

Dear Reviewer 2,

thank you very much for your valuable and comprehensive feedback on our manuscript which we appreciate a lot. We have reworked the manuscript and hope that the adaptations will meet your criticism. We revised the manuscript with tracked changes to highlight the revisions. The lines indicated refer to the new lines in the revised manuscript. Please find below a point-by-point response to your comments and suggestions.

Comment 1

Introduction - Strengths:

Contextual Relevance: The inclusion of current global drivers like digitalization, ecological transformation, and demographic changes is highly relevant. The connection to how these factors impacts work environments and psychosocial risks is timely and significant.

Clear Problem Statement: The problem you aim to address is clearly articulated: the relationship between organizational responses to the pandemic and employees' perceived psychosocial demands. This is an important and under-explored area in occupational health and safety research.

Comprehensive Literature Review: You effectively refer to prior research, including the work of the ILO, OECD, and previous crises like the Ebola epidemic and the 2008 economic recession, which supports the urgency of understanding psychosocial risks in the workplace. The inclusion of both physical and psychosocial demands aligns well with the focus of your paper.

Use of Case Examples: Your examples, such as the changes in working conditions during the COVID-19 pandemic (e.g., remote work, social distancing), help to ground the theoretical framework in real-world changes. This provides practical relevance and context to your research.

Methodological Approach: The combination of qualitative and quantitative methods adds depth to your investigation. It seems like you're trying to bridge a knowledge gap by exploring both employee and manager perceptions during the pandemic.

Suggestions for Improvement:

Clarity and Flow: The first few sentences could benefit from more clarity. For example, the initial reference to ILO and OECD could be better connected to the rest of the paragraph to avoid abruptness.

"According to the International Labor Organization (ILO) and the Organization for Economic Co-operation and Development (OECD), increasing digitalization, ecological transformation, and advancing demographic change are key drivers of accelerated technological development and the constantly changing world of work."

This sentence could be shortened and made more direct for easier readability, such as:

"The International Labor Organization (ILO) and the Organization for Economic Co-operation and Development (OECD) highlight that digitalization, ecological transformation, and demographic changes are reshaping the world of work."

Answer 1

Thank you for highlighting the importance of the research to

---

## [Editor Report · Decision Letter 2]

26 Jun 2025

Dear Dr. Soeder,

Thank you for submitting your manuscript to PLOS ONE. After careful consideration, we feel that it has merit but does not fully meet PLOS ONE’s publication criteria as it currently stands. Therefore, we invite you to submit a revised version of the manuscript that addresses the points raised during the review process.

I have carefully reviewed your responses to the reviewers' comments, as well as the updated version of the manuscript. I would like to commend you on the thoughtful and thorough way in which you addressed the points raised during the review process. The revisions you have made significantly enhance the clarity and overall quality of the work. In my assessment, your responses adequately and satisfactorily resolve most of the reviewers’ concerns.

Based on my reading of the manuscript, I believe that it has the potential to be published in PLOS ONE, provided that a few remaining minor issues are addressed. I therefore invite you to submit a further revised version of the manuscript that takes these final points into account (please see below). Once the revised manuscript is received and assessed, we will be in a position to move forward with a final decision.

https://journals.plos.org/plosone/s/submission-guidelines#loc-laboratory-protocols . Additionally, PLOS ONE offers an option for publishing peer-reviewed Lab Protocol articles, which describe protocols hosted on protocols.io. Read more information on sharing protocols at https://plos.org/protocols?utm_medium=editorial-email&utm_source=authorletters&utm_campaign=protocols .

We look forward to receiving your revised manuscript.

Kind regards,

Damiano GIRARDI

Academic Editor

PLOS ONE

Journal Requirements:

**Reviewers' comments:**

- p.2, line 21, "This longitudinal mixed-methods study explores how organizations dealt with COVID-19 pandemic-related challenges". The investigation involved workers employed within a single company group, which may limit the generalizability of the findings.

- p.4, line 74, "depending on the individual’s preconditions". Why do the authors mean by individual preconditions? Personality traits, socio-demographic characteristics, ... An example should clarify this point.

- p.7, line 136. This section appears somewhat heterogeneous, as it presents various types of information. In my opinion, the description of the study aims (e.g., 'findings from the interviews further helped to address how workers from different occupational fields experienced...'), the methodology (i.e., the mixed-method approach), and potentially the study’s contribution to the field, might be better organized under a dedicated section—for example, titled "The Current Study".

- p.21. The authors should clearly acknowledge that, in some cases, the number of participants within specific occupational fields was too small to support statistical inference or allow for generalization of the results.

---

## [Author Response · Author response to Decision Letter 3]

29 Jun 2025

Dear Prof. Dr. Damiano Girardi,

thank you very much for having taken the time and effort to carefully read through our manuscript and the previous reviewers’ comments. Thank you for your positive and encouraging feedback on our revised manuscript and responses to the reviewers' comments. We are pleased to hear that the revisions have improved the clarity and quality of the work and that most concerns have been satisfactorily addressed. We appreciate your suggestions on the remaining minor issues.

Please find below our answers to the remaining points on our manuscript now titled ‘Pandemic preparedness in shaping psychosocial working conditions – insights for occupational safety and health from a longitudinal mixed-methods study during the COVID-19 pandemic at six company sites of one organization in Germany’.

In addition, we uploaded a marked-up copy of our manuscript to highlight the changes made, labeled 'Revised Manuscript with Track Changes' as well as an unmarked version of the revised manuscript labeled ‘Manuscript’.

Thank you for having taken over this review process. We look forward to hearing from you.

Kind regards on behalf of all authors,

Corresponding author

Comment 1

p.2, line 21, "This longitudinal mixed-methods study explores how organizations dealt with COVID-19 pandemic-related challenges". The investigation involved workers employed within a single company group, which may limit the generalizability of the findings.

Answer 1

Many thanks for your recommendation. We have revised the sentence describing the objective of our study to avoid creating the false impression that the results can be generalized without limitation.

Changes in the manuscript

Abstract: This longitudinal mixed-methods study explores how one company group dealt with COVID-19 pandemic-related challenges and how employees and managers perceived work-related psychosocial demands in Germany. […] (lines 21-23) 

Comment 2

p.4, line 74, "depending on the individual’s preconditions". Why do the authors mean by individual preconditions? Personality traits, socio-demographic characteristics, ... An example should clarify this point.

Answer 2

Many thanks for that comment. We can see your point and agree. We now added three examples to clarify that point.

Changes in the manuscript

Introduction: Psychosocial demands can – depending on the individual’s preconditions such as socio-demographic characteristics, experiences, and current mental state – lead to either favorable (e.g., activating state) or unfavorable strain (e.g., fatigue or problems concentrating) and thus impact the individual’s health [7]. (lines 74-78) 

Comment 3

p.7, line 136. This section appears somewhat heterogeneous, as it presents various types of information. In my opinion, the description of the study aims (e.g., 'findings from the interviews further helped to address how workers from different occupational fields experienced...'), the methodology (i.e., the mixed-method approach), and potentially the study’s contribution to the field, might be better organized under a dedicated section—for example, titled "The Current Study".

Answer 3

Thank you for that comment. You are right, the section had become difficult to read. We have now restructured it and hope that the different types of information are now presented in a clearer and more understandable way.

Changes in the manuscript

Material and methods:

Study design

This study is part of a larger, explorative, modular study project that examines the acceptance of SARS-CoV-2-infection control measures in occupational settings (blinded reference). Ethical approval was obtained. The study project was conducted in accordance with the Declaration of Helsinki and relevant institutional guidelines.

For the current study, we applied a mixed-methods approach, commonly used in organizational health services research to consider the complexity of organizations [12, 13]. Using the combination of qualitative interviews and a quantitative survey, we aimed to gain insights into organizational pandemic preparedness in work design. Quantitative survey data were collected at multiple time points to monitor changes in how the employees’ and managers’ perceived psychosocial demands at work. Findings from the interviews further helped to address how workers of different working fields experienced support from both the German state (e.g., financial aid in the form of temporary short-time work compensation) and their organization (e.g., adjusted work arrangements) during the pandemic. These pluralistic perspectives enable an evaluation of how the organization’s responses to the challenges of the pandemic affected the employees’ daily work.

The findings from this study could help design more favorable and sustainable psychosocial working conditions in organizations that are similar to the one investigated. Designing working conditions based on the learnings from the COVID-19 pandemic can help ensure workability during future challenging times.

Mixed-methods approach

For contextualization, we performed a comprehensive document analysis of prevailing national and federal laws and OSH-regulations for infection control in Germany. In a parallel approach, quantitative (employee survey) and qualitative data (interviews with managers) were gathered and analyzed independently. This allowed us to gain detailed insights into the company-level context in which employees and managers work. Data integration was realized during sampling, as both, quantitative and qualitative data, were collected via convenience sampling from the same working fields to ensure comparability. Further, integration was realized during the exploratory analysis and interpretation phase. The triangulation of methods allowed us to investigate consistent patterns in perceived psychosocial demands within working fields as well as between employees and managers. The contextual insights, mainly provided by the qualitative data, allowed a more nuanced understanding of these patterns. We report following the GRAMMS-guidelines [14].

Study participants

The longitudinal data comes from employees and managers from various working fields from a worldwide leading global supplier of technology and services. The company group can be characterized as financially well-established and having a lot of human resources (blinded reference). For example, regarding the work environment, digitalized solutions and proven communication channels were already used before the COVID-19 pandemic. All employees and managers included were employed at six German company sites in the federal states of Bavaria, Baden-Wurttemberg, and Lower Saxony. Employees for example worked at the assembly line/production, in the office, company medical service, or factory security service of the same company group. We aim to differentiate between those working fields as they show large differences in how working conditions were adapted to ensure as-safe-as-possible workplaces. While office workers were mainly affected by remote work and online meetings where possible, assembly line/manufacturing workers experienced redesigned working conditions to allow for keeping a minimum of 1.5m distance between work stations, frequent cleaning of work equipment, and decoupled break times. Company medical service personnel in comparison were responsible for implementing SARS-CoV-2-infection control measures as well as ensuring testing and vaccination possibilities. All participants were at least 18 years old, had a German language proficiency of at least B1, and provided written informed consent.

Employees were eligible as interview partners if they had been employed for at least six months and were either member of the crisis management team or held an operational management position. No explicit exclusion criteria were defined beyond not meeting the pre-defined inclusion criteria. Due to the limited timeframe between the onset of the pandemic and the first wave of data collection, the employee survey was administered solely online. As a consequence, there may have been systematic exclusion of individuals without sufficient time or resources to participate such as internet or email access at work. This was previously discussed in (limited reference). Qualitative data were pseudonymized and quantitative data were anonymized prior to analysis. (lines 119-202) 

Comment 4

p.21. The authors should clearly acknowledge that, in some cases, the number of participants within specific occupational fields was too small to support statistical inference or allow for generalization of the results.

Answer 4

We added a statement to the 'Limitations' section to acknowledge that the small sample size limits the generalizability of the results.

Changes in the manuscript

Discussion - limitations: […] Office and assembly line work areas generally represent the largest groups of employees within a company, whereas groups such as the company medical service are naturally smaller. However, the sample size in some work areas was too small to draw statistical conclusions or generalize the results. (lines 598-602)

---

## [Editor Report · Decision Letter 3]

1 Jul 2025

Pandemic preparedness in shaping psychosocial working conditions – insights for occupational safety and health from a longitudinal mixed-methods study during the COVID-19 pandemic at six company sites of one organization in Germany

PONE-D-25-00343R3

Dear Dr. Soeder,

We’re pleased to inform you that your manuscript has been judged scientifically suitable for publication and will be formally accepted for publication once it meets all outstanding technical requirements.

Kind regards,

Damiano GIRARDI

Academic Editor

PLOS ONE

Additional Editor Comments (optional):

Dear Dr. Soeder,

Thank you for submitting the revised version of your manuscript to PLOS ONE. I am pleased to inform you that the revisions have satisfactorily addressed the concerns raised during the previous rounds of review.

The manuscript now meets the journal's standards for scientific rigor and clarity. Therefore, no further changes are necessary, and I am happy to inform you that your manuscript can be accepted for publication.

Congratulations on a fine piece of work.

Best regards,

Damiano Girardi
---

## [Editor Report · Acceptance letter]

PONE-D-25-00343R3

PLOS ONE

Dear Dr. Soeder,

I'm pleased to inform you that your manuscript has been deemed suitable for publication in PLOS ONE. Congratulations! Your manuscript is now being handed over to our production team.

Kind regards,

on behalf of

Dr. Damiano GIRARDI

Academic Editor

PLOS ONE